# PHOTOARTAGENT:
# INTELLIGENT PHOTO RETOUCHING WITH LANGUAGE MODEL-BASED ARTIST AGENTS

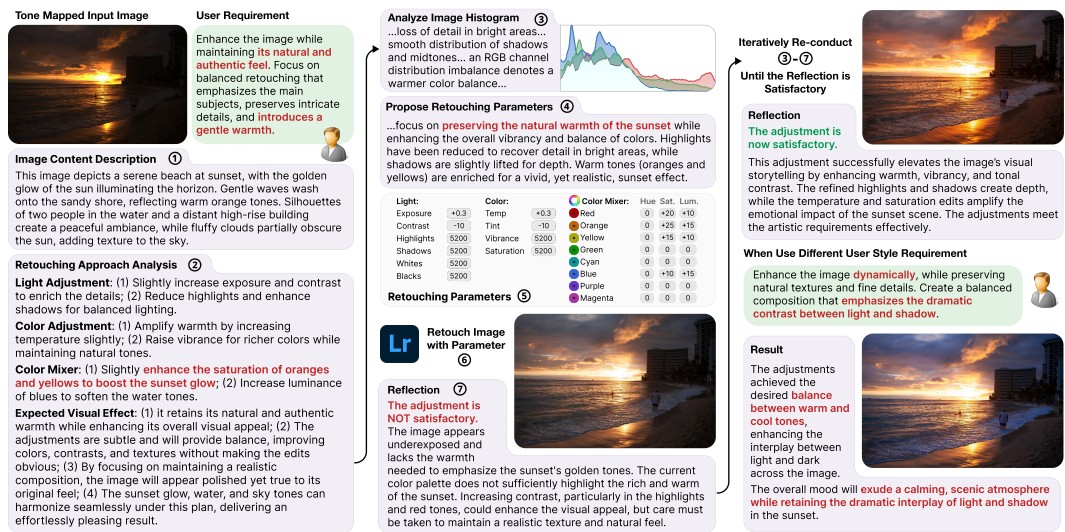

Figure 1: Illustration of PhotoArtAgent's workflow for photo retouching. Our system ① analyzes image content, ② proposes retouching strategy and art concept, ③ analyzes histogram, ④ and ⑤ generates retouching parameter, ⑥ applies parameter using the Lightroom software, and ⑦ conducts reflection for iterative refinement. PhotoArtAgent is able to change strategies based on user descriptions, retouching to completely styles. Text in light violet boxes represents PhotoArtAgent's outputs, while green boxes indicate user inputs. Key content is highlighted for emphasis.

## ABSTRACT

Photo retouching is integral to photographic art, extending far beyond simple technical fixes to heighten emotional expression and narrative depth. While artists leverage expertise to create unique visual effects through deliberate adjustments, non-professional users often rely on automated tools that produce visually pleasing results but lack interpretative depth and interactive transparency. In this paper, we introduce PhotoArtAgent, an intelligent system that combines Vision-Language Models (VLMs) with advanced natural language reasoning to emulate the creative process of a professional artist. The agent performs explicit artistic analysis, plans retouching strategies, and outputs precise parameters to Lightroom through an API. It then evaluates the resulting images and iteratively refines them until the desired artistic vision is achieved. Throughout this process, PhotoArtAgent provides transparent, text-based explanations of its creative rationale, fostering meaningful interaction and user control. Experimental results show that PhotoArtAgent not only surpasses existing automated tools in user studies but also achieves results comparable to professional human artists.

## 1 INTRODUCTION

Photo retouching, as a component of photographic art, extends beyond mere technical image defect correction to enhance the emotional expression and narrative value of photographic works. Artists combine various fundamental image processing operations to create unique visual tension through artistic adjustments in color and tonality. These seemingly simple operations can produce remarkable artistic effects that transcend their basic nature. To untrained observers, the photo retouching process may appear arbitrary and incomprehensible. However, each retouching decision stems from the artist's carefully considered creative intent. In the artist's perspective, every photograph harbors unique artistic potential. When confronted with different images, artists can envision clear artistic goals and, drawing upon their professional knowledge and rich experience, execute precise retouching operations to realize these visions, thereby maximizing the work's inherent value.

While ordinary users aspire to achieve their personal aesthetic goals through photo retouching, mastering professional retouching skills requires extensive training and experience. In response, numerous automated photo retouching tools have been developed, aiming to provide non-professional users with professional-grade editing capabilities. However, users often find themselves passively accepting algorithmic outcomes. Although some results may be visually satisfactory, existing automated tools lack the interpretative depth and unique perspective of an artist, limiting meaningful artistic interaction.

We posit that description and explanation are also crucial elements in artistic creation. The artist's conceptual interpretation, expressive direction, and work analysis form inseparable components of the artistic piece. Through such description and interaction, users can deepen their understanding of the work and gain greater creative control. We hope to have a retouching system that can explicitly express the understanding of the image and retouch through reasoning. The system not only integrates professional photo editing expertise, gradually realizes artistic concepts through basic operations, but also explains the motivation and reasoning behind each step, achieving complete transparency. In addition, the system can also accept user input through text descriptions or reference images, further enriching its functions through multi-modal capabilities.

However, implementing such a system poses significant challenges. Existing methods primarily learn implicit patterns from training data. We face difficulties in interpreting the outputs of these methods, let alone providing explanatory descriptions alongside their predictions. Even more challenging is our goal of enabling models to assess artistic value and conceptualize creative directions in photography. While some approaches attempt to emulate human artists' tools, their decision-making processes for tool selection remain implicit and difficult to explain. These methods generally lack the capability for intelligent interaction with users. To advance automated photo retouching into a new era, we need a system that fundamentally departs from current paradigms.

In recent years, Large Language Models (LLMs) have made transformative strides in natural language understanding and generation, showcasing impressive reasoning abilities and knowledge integration to handle complex contexts and produce coherent descriptions. Vision-Language Models (VLMs) have also advanced significantly in image understanding and cross-modal reasoning. Notably, these models show promising potential in understanding artistic features of images and generating aesthetic descriptions. Language provides users with a direct and intuitive means of human-computer interaction. Furthermore, LLM agents offer new possibilities for executing complex photo retouching operations, as they can interpret user intent, plan execution steps, and provide clear explanations. These technological advancements create opportunities for photo retouching systems. By integrating the reasoning capabilities of LLMs and the visual comprehension abilities of VLMs, we can design an agentic system that both understands the artistic creative process and engages in natural interaction with users. Such a system can not only "perceive" the artistic potential within photographs but also articulate creative intentions in natural language and explain retouching steps, emulating the creative process of human artists.

In this paper, we develop an intelligent agent named PhotoArtAgent, which leverages VLMs for active reasoning while employing photo editing software to refine photographs. PhotoArtAgent is capable of conducting explicit artistic analyses on the target image and conceptualizing creative directions based on flexible user instructions and requirements. It then outputs the specific parameters needed for retouching. Through an API interface, the system collaborates with Lightroom to apply these parameters. Subsequently, PhotoArtAgent evaluates the resulting images to determine

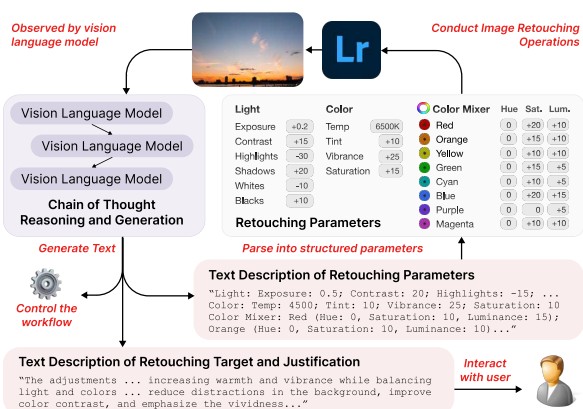

Figure 2: Overall schematic of the core paradigm of our methodology. Our approach utilizes chain-of-thought reasoning from a VLM to understand both the image and user requirements, generating textual descriptions. These descriptions convey feedback to the user, control signals for the workflow controlling and specify the retouching parameters. The parameters are used to control the Lightroom software. The VLM then re-examines the processed image, completing one iteration of the reflection loop.

whether they fulfill the intended artistic vision. If the outcome remains unsatisfactory, the agent proposes new parameters, refining the photo step by step by building on insights from previous attempts. Throughout each stage of this process, PhotoArtAgent communicates with users through textual explanations, clarifying its creative concepts and operational rationale. We show an illustration of the outputs and workflow of the proposed system in Figure 1. Our experiments show that PhotoArtAgent not only accomplishes the proposed objectives but also outperforms existing methods in user studies, achieving results even comparable to those of professional human artists. Beyond expanding the possibilities of photo retouching, we also offer new insights on how artificial intelligence can support artistic creation.

## 2 PHOTOARTAGENT

We present PhotoArtAgent, an automated, training-free system for artistic photo retouching. Figure 2 illustrates the core logic of the methodology behind the proposed PhotoArtAgent. To enable the VLM to perform retouching operations, we build a tool interface that allows it to call external tools (Sec 2.1). Building on this, we developed a workflow to ensure the efficient collaboration of VLMs across different stages (Sec 2.2). Finally, we demonstrate the interaction process of PhotoArtAgent and how to customize the model through an external knowledge base (Sec 2.3).

### 2.1 RETOUCHING TOOLBOX

Photo retouching methods fall into two categories: end-to-end image models or filters (Chen et al., 2018b;a), and interpretable operations with explicit physical meanings (Hu et al., 2018; Tseng et al., 2022). The second category is more user-friendly, as the operations and tools have explicit physical meanings, allowing users to flexibly choose and adjust retouching results. Common photo editing toolkits include Lightroom [1], Darktable [2], etc. In this study, we implement an API to operate Lightroom, allowing the proposed PhotoArtAgent system to use various Lightroom operations.

There are several reasons for choosing Lightroom as the core toolbox: (1), Lightroom is widely used as a professional photo editing software, with abundant discussions and tutorials available online. These resources form part of the training data for many language models, equipping them with a substantial understanding of Lightroom operations. Leveraging Lightroom allows us to capitalize on this advantage; (2), Lightroom's editing operations have been thoroughly tested and deliver consistently high-quality image enhancement results. Moreover, our method does not require the image operations to be differentiable, allowing direct use of Lightroom's built-in tools; (3), Lightroom provides an intuitive interface that is familiar to users. Our system interacts with Lightroom in a human-like way, ensuring that the editing process remains highly interpretable and transparent to users. They can further refine the retouched results generated by our system if desired.

In terms of specific operations, we incorporate three major adjustment categories from Lightroom:

---

[1] A professional photo editing software developed by Adobe Inc.

[2] An open-source photography workflow application

- **Light Adjustments**: This includes adjusting parameters such as `exposure`, `contrast`, `highlights`, `shadows`, `whites`, and `blacks`. These operations are crucial for overall tonal adjustments and light distribution in the image.
- **Color Adjustments**: This includes adjustments to `color temperature`, `tint`, `saturation`, and `vibrance`. These parameters can globally alter the image's color tone, affecting the warmth, softness, or vibrancy of the visual output.
- **Color Mixer**: This feature allows precise adjustments to eight specific color channels – `red`, `orange`, `yellow`, `green`, `cyan`, `blue`, `purple`, and `magenta` – in terms of `hue`, `saturation`, and `luminance`. It enables targeted modifications to enhance, suppress, or alter specific colors for artistic expression.

## 2.2 Cognitive Architecture/Workflow of Photo Retouching

PhotoArtAgent requires extensive knowledge and reasoning capabilities across multiple domains, including image analysis, photography, Lightroom operations, and artistic principles. These features are all supported by VLM through the generation of text. However, solving such complex tasks cannot be accomplished through a single forward inference of the language model. A common approach involves using a task-specific, chain-of-thought process to break down complex problems into multiple reasoning steps for step-by-step resolution (Wei et al., 2022; Yao et al., 2024). Inspired by the internal logic employed by human experts, we abstract the photo retouching task into a series of subtasks, each handled by sequential calls to the VLM. The design of these subtasks and their procedural flow is referred to as the cognitive architecture. In this section, we provide a detailed description of the cognitive architecture designed in our approach. A schematic diagram of the cognitive architecture is shown in Figure 3. Although our method divides the

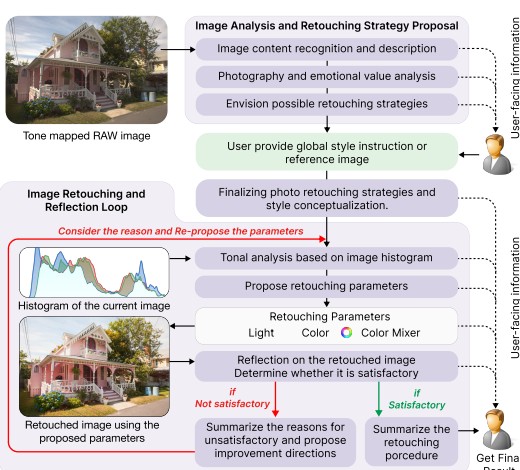

Figure 3: The diagram illustrates the PhotoArtAgent's workflow, divided into two main components: Image Analysis and Retouching Strategy Proposal (top) and Image Retouching and Reflection Loop (bottom).

photo retouching process into distinct steps, all these steps are performed by the same VLM with different prompts. To guide the VLM in fulfilling various functions, we designed specific requirements and prompts tailored to each step, ensuring that the model can effectively address the unique tasks of each phase. More details of the prompt design can be found in the supplementary materials.

**Overview** Our approach can be broadly divided into two parts. The first part, referred to as *Proposal for Image Analysis and Retouching Strategy*, focuses on analyzing and interpreting the image while interacting with the user to establish an overarching photo retouching goal. This mirrors the initial task of human artists during artistic creation. When faced with a new photograph, human artists rely on their expertise and artistic vision, combined with the user's requirements, to propose a general direction and vision for the creative process. The second part, known as the *Image Retouching and Reflection Loop*, employs an iterative workflow of retouching and reflection to refine the image. This process also aligns closely with the behavior of human artists during creation. Given the complexity and multitude of parameters involved in photo retouching, achieving the desired result through a single prediction is often impractical. Human artists typically engage in multiple trials, drawing inspiration and direction from the outcomes of different iterations to progressively improve the work.

**Proposal for Image Analysis and Retouching Strategy.** In this initial stage, PhotoArtAgent performs image content recognition to identify the main subject, background, and compositional elements. It then provides a photography and emotional value analysis, assessing factors such as the mood conveyed by the scene, the style or genre, and any relevant artistic cues. Building on

this analysis, the system envisions possible retouching strategies and presents them to the user in a high-level format. These strategies can range from enhancing tonal balance and contrast to establishing a vintage or modern aesthetic, depending on user preferences. During this process, the user may provide a global style instruction or reference image to guide PhotoArtAgent's retouching approach. With these instructions in mind, PhotoArtAgent then finalizes photo retouching strategies and conceptualizes a coherent style that suits both the user's direction and the inherent qualities of the photograph. By the end of this stage, the system has produced a strategic blueprint that forms the basis for detailed editing parameters in the next phase.

**Retouching Parameter Generation.**  Next, the system transitions from abstract strategy formulation to concrete adjustments by proposing specific photo retouching parameters. First, histogram images are provided to the VLM, which performs tone analysis. By analyzing the distribution of pixel values across different color channels, PhotoArtAgent identifies potential issues such as overexposure, underexposure, excessive contrast, or color imbalance. Based on the tone analysis, the system generates retouching parameters for the current image along with explanations for the suggested adjustments, as shown in Figure 2. Leveraging the function calling feature of advanced VLMs (Patil et al., 2023), these parameters are output in a structured JSON format. The structured parameters are directly mapped to the settings of Lightroom, where the retouching operations are applied to the image, ensuring a smooth transition from analysis to actionable operations.

**Reflection**  After applying the suggested parameters to the image, PhotoArtAgent enters the reflection phase. Due to the inherent complexity of interactions between photo retouching operations and the image itself, it is challenging for any approach to determine the optimal parameters directly from the original image. Observing the results and making incremental adjustments is an intuitive process, commonly employed by human artists. In this phase, the system analyzes the retouched image to assess whether the modifications align with the user's goals. If the results are deemed unsatisfactory, PhotoArtAgent identifies the reasons behind the shortcomings and proposes directions for improvement. Rather than directly attempting to refine the retouching parameters, identifying areas of dissatisfaction and targeting specific adjustments is a much more manageable task. The system then loops back to generate new editing parameters. If the image meets or exceeds expectations, PhotoArtAgent concludes the process by summarizing the editing workflow, providing a concise explanation of how the adjustments influenced the final image. This reflection-driven loop ensures that the retouching process remains flexible and responsive to feedback.

## 2.3 INTERACTION

The cognitive architecture described above establishes a foundational framework for utilizing VLMs in photo retouching. This framework offers significant scalability and flexibility. In the following sections, we describe PhotoArtAgent's interactive capabilities and its ability to perform photo retouching based on complex and even multi-modal user instructions.

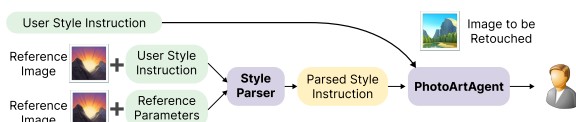

Figure 4: Overview of PhotoArtAgent's flexible input interface. The workflow supports multi-modal interactions including text instructions, reference images, and cases with retouching parameters.

**Explanatory Information Output.**  The proposed PhotoArtAgent system stands out from other automatic photo retouching systems partially due to its proactive reasoning capabilities and text-based output format. PhotoArtAgent not only performs explicit aesthetic reasoning but also presents its thought processes in a manner that is understandable to users. At each stage of its operation, the system provides textual feedback reflecting its intermediate reasoning, allowing users to comprehend the system's editing approach and direction. This feature grants PhotoArtAgent significant advantages over existing solutions. Figure 11 illustrates a photo retouching workflow, while Figure 1, Figure 8, Figure 9, and Figure 10 all showcase descriptive statements generated by the system. These descriptions enable users to understand how the system interprets the photo, uncovering details and highlights they might have overlooked. Additionally, users can grasp the rationale behind the system's specific editing operations. Since PhotoArtAgent directly operates Lightroom software

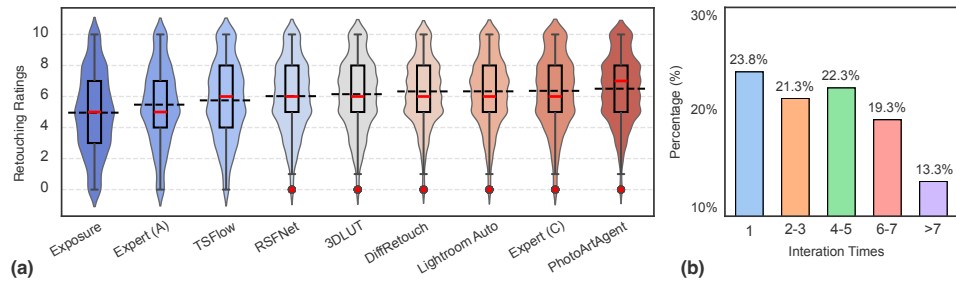

**(a)** **(b)**

Figure 5: (a) Comparison of retouching ratings across different methods. The violin plots show the distribution of ratings from 0–10, with box plots indicating quartile ranges. (b) Analysis of the reflection frequency, demonstrating that 76.2% of images required multiple iterations to achieve optimal results.

to perform image adjustments, users can further refine the edits based on the system's outputs. With the system providing well-explained reasoning for its suggestions, users can make targeted modifications to achieve their desired outcomes. These features endow PhotoArtAgent with a level of intelligence and interactivity far surpassing existing approaches, bridging the gap between automated editing and human creativity.

**Multi-modal User Instruction.** PhotoArtAgent accepts user-provided style instructions as input. While the system can proactively analyze an image and suggest artistic concepts, the final retouching strategy depends on user guidance. Users may provide textual goals, reference images, or retouching parameters. For instance, as shown in Figure 4, a style parser powered by VLMs can interpret a reference image into textual descriptions for application. This flexible design enables diverse human–computer interaction and supports varied creative workflows.

## 3 RESULTS

**Implementation** We conducted our experiments using Adobe Lightroom Version 8.1. In terms of the VLM, we required both function-calling capabilities and multi-modal support, so we primarily relied on commercial solutions. We explored three multi-modal language models in our research: GPT-4o (`gpt-4o-2024-11-20`), GPT-4o-mini (`gpt-4o-mini-2024-07-18`), and Claude 3.5 Sonnet (`claude-3-5-sonnet-20241022`). User interactions and information displays were facilitated via a web-based chat interface.

**Comparison with Existing Methods** Our primary experiments were conducted on the MIT-Adobe FiveK dataset (Bychkovsky et al., 2011). We first compare our approach against several existing methods, including deep models RSFNet (Ouyang et al., 2023) and TS-Flow (Wang et al., 2023), 3D lookup table method 3DLUT (Zeng et al., 2020), reinforcement learning-based method (Hu et al., 2018), and recently proposed diffusion model-based solution (Duan et al., 2024). We also include the auto retouching feature in Lightroom software, as well as the results from two human experts. Because our method is training-free, there was no need to split the data into separate training and testing sets; instead, we randomly sampled 115 images for systematic comparison.

Table 1: Quantitative comparison of different photo retouching methods based on User Study ratings (1-10) and GPT-4V evaluation scores.

| Method | User Study | GPT-4V Score |
|---|---|---|
| Exposure | 4.96 | 5.67 |
| TSFlow | 5.75 | 6.05 |
| RSFNet | 6.02 | 5.86 |
| 3DLUT | 6.15 | 6.11 |
| DiffRetouch | 6.32 | 5.95 |
| Lightroom Auto | 6.36 | 5.89 |
| Human Expert (A) | 5.47 | 5.90 |
| Human Expert (C) | 6.33 | 6.10 |
| PhotoArtAgent (ours) | **6.50** | **6.17** |

Evaluating photo retouching is inherently challenging, as even human experts may differ on the "best" outcome. Consequently, reference-based metrics are often not appropriate. Instead, we conducted an extensive user study, asking participants to score each retouched image in terms of quality and style on a scale of 0 (noticeable flaws) to 10 (creative and satisfying). A total of 125 volunteers

– recruited through the Prolific platform and required to have photography experience – participated in this evaluation. Each image thus received 125 separate ratings. In this user study, we focused only on retouching results rather than any interactive features or user experience improvements.

Table 1 presents the average scores for all compared methods. Our approach achieves significantly higher user satisfaction than prior methods and even outperforms the commercial Lightroom Auto feature. The comparison with the two human experts yielded surprising results: the most advanced automated retouching systems closely match Expert C's. Notably, PhotoArtAgent obtained an even higher score than both human experts, suggesting that agent-based retouching techniques hold tremendous potential. Violin plots illustrating the score distributions are shown in Figure 5 (a), indicating that users view PhotoArtAgent as more creative. Qualitative comparisons of selected images are provided in Figure 7. Lastly, we also provide aesthetic evaluation results using GPT as an additional reference in Table 1. The application of VLMs for assessing image quality and aesthetics has become increasingly prevalent (Jia et al., 2023; Inoue et al., 2024; You et al., 2025; 2024; Wu et al., 2023b; Chen et al., 2024b; 2025). In certain scenarios, such tools can serve as an additional benchmark or complementary perspective. Our method also has an advantage in this indicator.

**Effects of the Reflection Mechanism**  We next examine the necessity of several key design elements idn PhotoArtAgent, focusing first on the reflection mechanism. Our system relies on this iterative reflection process to attain more refined results. Figure 8 illustrates how reflection operates by showcasing sample textual outputs from the system. In the example, the system initially underestimates the exposure adjustment. In the reflection stage, PhotoArtAgent increases the exposure by 0.2 EV, overbrightening the image. It then corrects this by reducing the exposure for a more balanced result. Recognizing this issue, the system lowers the exposure again for a more balanced outcome. A similar pat-

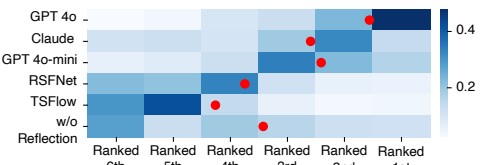

Figure 6: Ranking distribution of different photo retouching configurations, where the color intensity represents the proportion of rankings received (darker blue indicates higher percentage), and red dots indicate the mean ranking position for each method.

tern is also observed with color temperature adjustments. In the second example, PhotoArtAgent also raises the exposure in multiple steps: the first attempt remains insufficient, prompting the system to increase exposure further. These examples further demonstrate the system's ability to modify color to achieve specific objectives – details that can be gleaned from the provided textual explanations. We encourage readers to review these explanations closely, as they highlight PhotoArtAgent's intelligent decision-making process. We also investigated how many images in our trials benefited from the reflection process. Figure 5 (b) presents the distribution of iteration counts across more than 1,000 runs in our experiment, revealing that only 23.8% of the images achieved satisfactory results on the first attempt.

**Ablation User Study**  We performed an ablation user study to assess the key components of our method, specifically the Reflection mechanism and different VLMs. Users were asked to rank the results from six different configurations. As shown in Figure 6, our method without the Reflection mechanism did not demonstrate a clear advantage over RSFNet or TSFlow. Once Reflection was integrated, our approach consistently outperformed the other methods. Among the tested VLMs, GPT-4o-mini and Claude underperformed compared to GPT-4o, possibly due to GPT-4o-mini's reduced model size and the fact that our prompts were designed based on GPT-4o. Nevertheless, these results underscore the potential of our proposed system paradigm.

**Effects of Different User Instructions**  Our method can retouch photos according to a wide range of user requirements. It processes and interprets user demands without requiring training, endowing the system with remarkable adaptability. Figure 9 illustrates two examples. In the example on the left, when users request a natural retouch, PhotoArtAgent applies typical exposure and color adjustments. However, if the user specifies a vintage black-and-white style, PhotoArtAgent not only understands this instruction but also infers that reducing color saturation to "-100" will achieve the desired monochrome effect. In the example on the right, the agent tailors distinct retouching strategies based on diverse stylistic or mood descriptions, generating multiple sets of results with distinctly different colors and atmospheres, all at a consistently high standard.

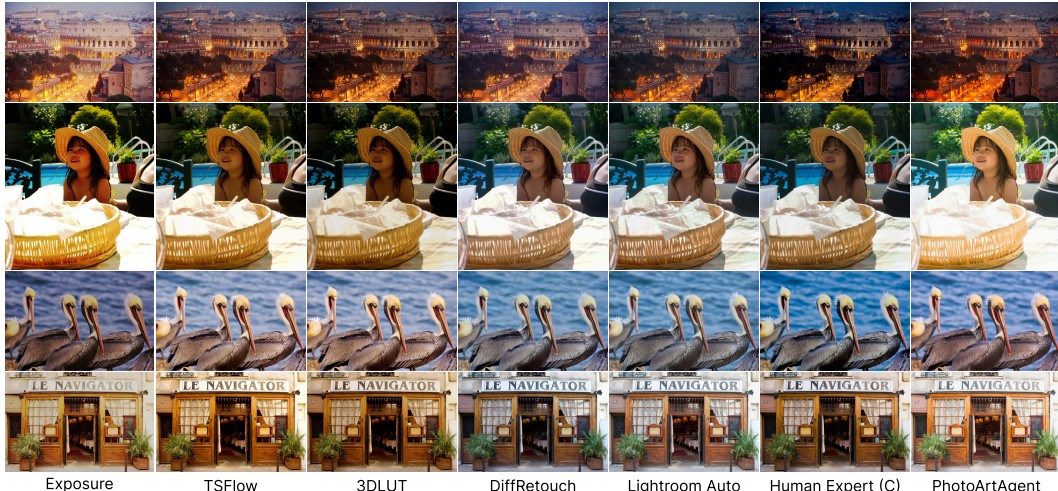

Exposure     TSFlow     3DLUT     DiffRetouch     Lightroom Auto     Human Expert (C)     PhotoArtAgent

Figure 7: Visual comparison of retouching results across different methods.

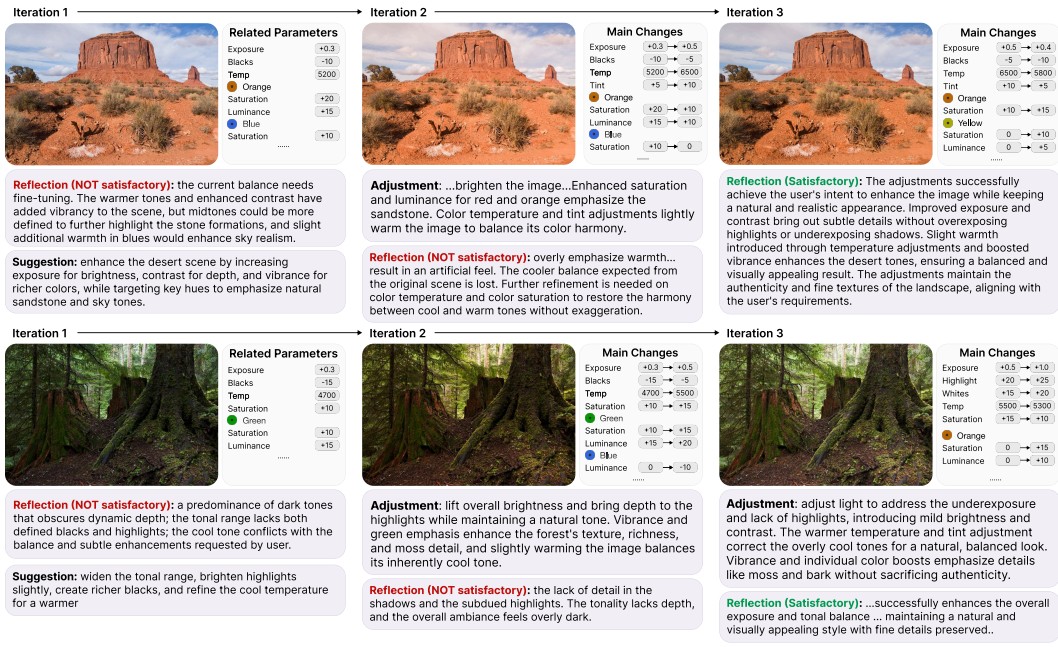

Figure 8: Visualization of PhotoArtAgent's iterative reflection process on two different scenes. Each iteration includes the applied retouching parameters and corresponding reflection text that guides subsequent refinements until achieving satisfactory results.

PhotoArtAgent can also accept multi-modal user inputs as user instructions. A dedicated VLM, serving as a "style parser," converts these multi-modal inputs into text-based instructions for PhotoArtAgent. Figure 9 presents two cases. In the first case, when users supply two reference images representing a desired style, the parser examines the stylistic features of both images and conveys the findings to PhotoArtAgent. Guided by this style information, PhotoArtAgent then retouches the target photo. As seen from the resulting images, the system accurately captures critical attributes to produce distinctly different yet stylistically faithful outcomes.

The second scenario highlights even greater potential. An underexposed photo with a large dynamic range initially fails to achieve a satisfactory result with the default approach. However, when provided with a similar high-contrast image along with its retouching process as a reference, PhotoArtAgent extracts useful insights to refine its retouching strategy, yielding a markedly improved result. Paired with a database of past examples, this capability could pave the way for a personal-

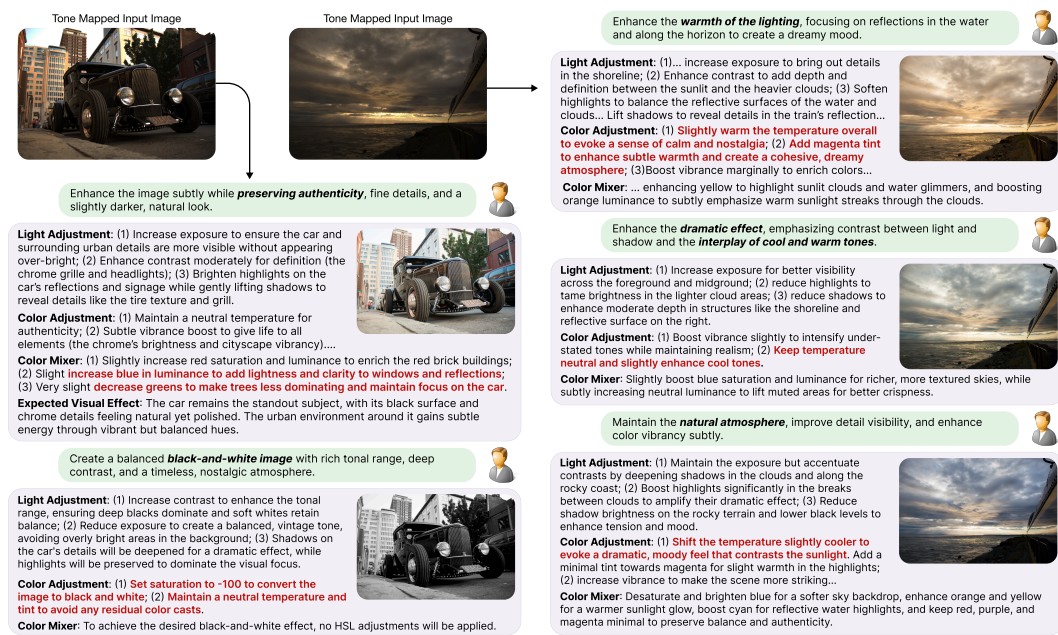

Figure 9: Examples showcase how PhotoArtAgent adapts its retouching instructions to meet user requirements: preserving authenticity in a car photo, rendering the same car in black-and-white, and enhancing various lighting and mood attributes in a seascape scene. Each case includes the input image alongside detailed light adjustments, color modifications, and mixer settings, illustrating PhotoArtAgent's versatile capabilities in fulfilling diverse artistic objectives.

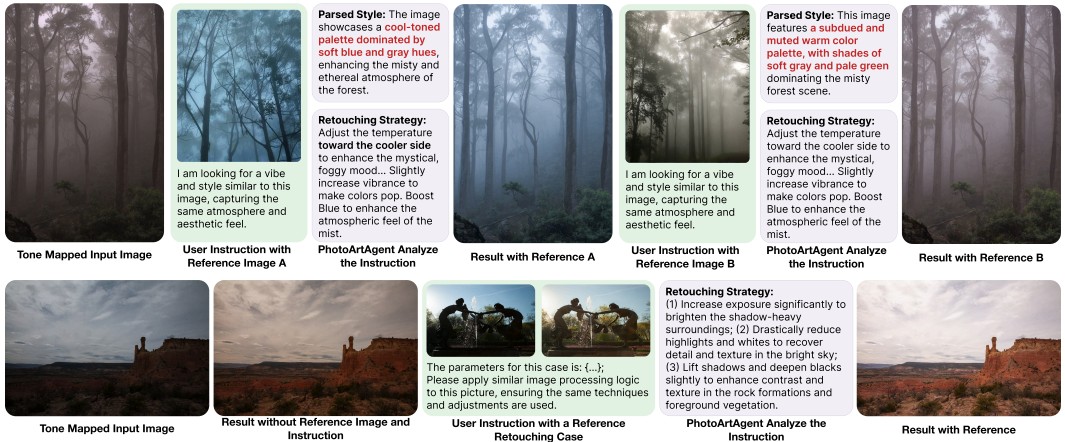

Figure 10: Demonstration of PhotoArtAgent's ability to handle diverse input prompts and reference images. Top: Two examples of forest scene retouching using different reference images, showing parsed style descriptions and corresponding results. Bottom: Desert landscape retouching comparing baseline output (without reference) against result guided by reference retouching parameters.

ized photo retouching assistant. We encourage readers to review the system's textual explanations in detail to gain a deeper appreciation of PhotoArtAgent's intelligence.

## 4 CONCLUSION

We present PhotoArtAgent, an intelligent photo retouching system that combines VLMs with agent-based reasoning to mimic a human artist's creative workflow. Through iterative parameter refinement, artistic evaluation, and transparent explanations, PhotoArtAgent tackles the challenges of interpretability and artistic vision in automated photo retouching.

## REPRODUCIBILITY STATEMENT

We emphasize reproducibility as a core principle of this study. The Appendix provides the complete descriptions of the prompts, while the main text presents a systematic account of the methods and evaluation protocols, with all assumptions and hyperparameters explicitly reported. To further facilitate verification and future research, we will release the analysis scripts, and the full source code on GitHub upon publication.

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

## A  RELATED WORK

**Photo Retouching**   Automated photo retouching methods can be categorized into end-to-end image processing models and physically inspired models. End-to-end models learn mappings from original to retouched images (Chen et al., 2018a;b; He et al., 2020; Kim et al., 2021; Deng et al., 2018; Kim et al., 2020; Zhu et al., 2017). They are often confined to patterns present in training datasets and lack interpretability (Gu & Dong, 2021; Shen et al., 2020; Gu et al., 2024). Physically inspired models frame image enhancement as a parameter optimization/learning problem for physical models (Liu et al., 2021; Wang et al., 2019; Zhu et al., 2020). These approaches include methods based on 3D look-up tables (Wang et al., 2021; Yang et al., 2022a; Zeng et al., 2020; Zhang et al., 2022; Yang et al., 2022b), parameterized filters (Hu et al., 2018; Tseng et al., 2022), and curve-based image adjustments (Li et al., 2020; 2022; 2021; Moran et al., 2021; Song et al., 2021; Gharbi et al., 2017; Moran et al., 2020; Chai et al., 2020). Most relevant to our work are approaches based on reinforcement learning (Hu et al., 2018; Kosugi & Yamasaki, 2020; Ouyang et al., 2023; Tseng et al., 2022; Ke et al., 2022). These methods also attempt to simulate human artists' photo retouching interactions, decomposing photo retouching into a series of operations and estimating their parameters. Although these methods formally mimic human workflows and offer some transparency in tool usage, they fail to capture the artistic vision inherent in human creative processes and lack deeper user interaction capabilities.

**Language Models and Vision Language Models**   Recent advances in Large Language Models (LLMs) have revolutionized natural language processing capabilities (Brown et al., 2020; Ouyang et al., 2022; Achiam et al., 2023; Touvron et al., 2023; Jiang et al., 2023; Dubey et al., 2024). These models, trained on vast text corpora, generate text through next-token prediction. The generated text exhibits knowledge and logical reasoning abilities learned from extensive corpora, demonstrating powerful potential in complex reasoning (Wei et al., 2022; Kojima et al., 2022), step-by-step planning (Yao et al., 2024; 2022), and domain-specific knowledge application (Muennighoff et al., 2023; Hazra et al., 2024). Vision language models (VLMs) further connect language generation with multi-modal visual inputs, enabling models to understand images (Radford et al., 2021) and generate text based on image inputs (Liu et al., 2024; Diao et al., 2022; Achiam et al., 2023; Driess et al., 2023; Koh et al., 2023; You et al., 2025; 2024). LLMs and VLMs are crucial for implementing the system described in this paper for three main reasons. First, these advanced models can engage in intelligent interaction with users through text generation. They can explain current operations and accept user instructions through dialogue. Second, language model training data contain extensive material on photographic aesthetics appreciation, and LLMs excel at leveraging existing corpora for creative output based on input images. Third, LLM-based agents have opened new possibilities for executing complex tasks, which we detail in subsequent sections.

**LLM Agents**   An agent is a program built to reach a goal by perceiving its surroundings and engaging with it through the tools at its disposal. These agents are autonomous and capable of operating independently without human intervention, while also adopting a proactive approach in pursuing their goals (Franklin & Graesser, 1996; Li et al., 2024; Xi et al., 2023). Early agent programs primarily relied on symbolic approaches (Franklin & Graesser, 1996) and reinforcement learning (Silver et al., 2018; Hu et al., 2018; Hwangbo et al., 2019). Recent years have witnessed transformative advances through LLM-based agent systems (Yao et al., 2022; Chen et al., 2023). Unlike traditional reinforcement learning agents, LLM-based agents can maintain explicit long-term plans and leverage extensive general knowledge for flexible problem-solving (Achiam et al., 2023). This capability has been likened to human cognitive processes, where LLMs' basic responses parallel System 1 (automatic thinking), while compound agent systems emulate System 2 (deliberate thinking) (Yao et al., 2024; Kahneman, 2011; Lin et al., 2024). Recent work has explored various agent architectures that enhance LLMs' problem-solving capabilities through structured mechanisms (Wang et al., 2024; Ridnik et al., 2024). Notable approaches include tree-based and graph-based search strategies (Yao et al., 2024; Besta et al., 2024), external tool integration (Shen et al., 2024; Wu et al., 2023a), memory retrieval systems (Zhu et al.; Park et al., 2023), and error-based learning mechanisms (Shinn et al., 2024; Yao et al., 2022). These agent systems demonstrate promising potential for complex task execution by combining LLMs' reasoning capabilities with structured problem-solving frameworks (Durante et al., 2024). There have also been some pioneering works that have tried to apply

the capabilities of LLM agents to solve complex image processing problems (Zhu et al., 2024; Chen et al., 2024a). The agent paradigm offers unique advantages for photo retouching tasks.

# B ADDITIONAL DISCUSSIONS

Due to space constraints in the main paper, we provide further discussion on several important aspects here.

## B.1 COST

Currently, PhotoArtAgent does not require any dedicated training. Its primary expense comes from calls to large-scale language models. For instance, in over 600 tests conducted using GPT-4o, the total API cost was about $68.41 – an average of $0.11 per image. While this is higher than purely end-to-end methods, it remains reasonable given the system's large-model foundation, interactive features, and demonstrated performance. Moving forward, the cost can be reduced by streamlining the cognitive architecture—keeping only essential information—and by setting limits on the number of reflection steps. Conversely, one could invest in more detailed cognitive processes to achieve greater intelligence at a higher cost.

## B.2 RUN TIME

Because our method interacts with Adobe Lightroom via a graphical user interface to perform image operations, a significant portion of the runtime is spent communicating with Lightroom. A typical run of PhotoArtAgent takes around two minutes: roughly one minute waiting for responses from the language model API, and another minute spent on Lightroom operations. We plan to improve this by employing faster APIs and parallelization strategies.

## B.3 CONSISTENCY

Language models operate by predicting the probability of each possible next token in a sequence. When generating text, these probabilities can be used in different ways, resulting in varying degrees of creativity and consistency. A common method is temperature scaling, where a parameter $\tau$ adjusts how "evenly" the model distributes probability among potential next tokens. A higher temperature makes the probability distribution flatter, increasing the likelihood of less probable tokens and thus introducing more variability. Conversely, a temperature of zero corresponds to greedy decoding, where at each step the model always selects the most probable next token. This results in deterministic, reproducible outputs for a given input. However, it also removes the serendipity that can be valuable in creative processes, where randomness can spark fresh ideas. Hence, the choice of temperature (and other sampling methods, like top-k or top-p sampling) offers a trade-off: reliability and reproducibility on one hand, and diversity and originality on the other. Figure 12 shows examples of this consistency issue.

## B.4 SCALABILITY

Our approach is highly extensible. Although we only explored light adjustments, color adjustments, and Color Mixer HSL adjustments in this study, any tool that supports programmatic access could be integrated. For example, additional Lightroom features could be controlled via the API, or PhotoArtAgent could call upon other automated photo-enhancement models for inspiration. In the future, PhotoArtAgent may even leverage internet resources – for instance, web searches – to further enrich its artistic process. Our work merely serves as an innovative demonstration of how a language-model-based agent can be used for photo retouching and creative tasks. As with building blocks, different language model calls can be combined flexibly to construct increasingly complex and intelligent systems, guided by a thoughtfully designed cognitive architecture.

## B.5 HALLUCINATION OF THE LANGUAGE MODEL

PhotoArtAgent relies heavily on the reasoning and judgment capabilities of large language models. In most situations, the model provides convincing, coherent outputs. However, we have also

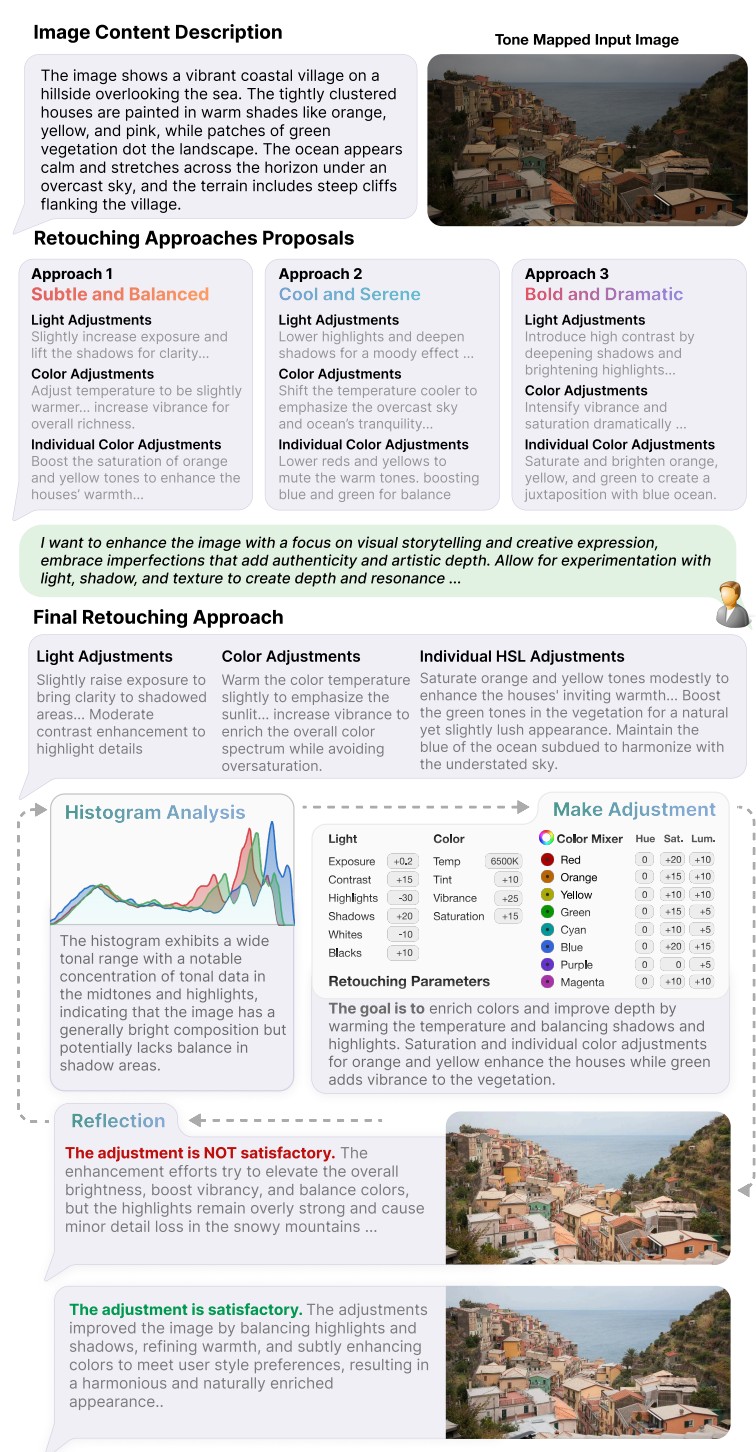

Figure 11: Comprehensive example of PhotoArtAgent's workflow, demonstrating the system's analysis and retouching process for a coastal village scene.

observed cases of hallucination – instances in which the model generates inaccurate or misleading responses. Hallucination occurs when the model invents information lacking any factual basis or evidence. Possible causes include flawed or insufficient training data and biases within the model itself.

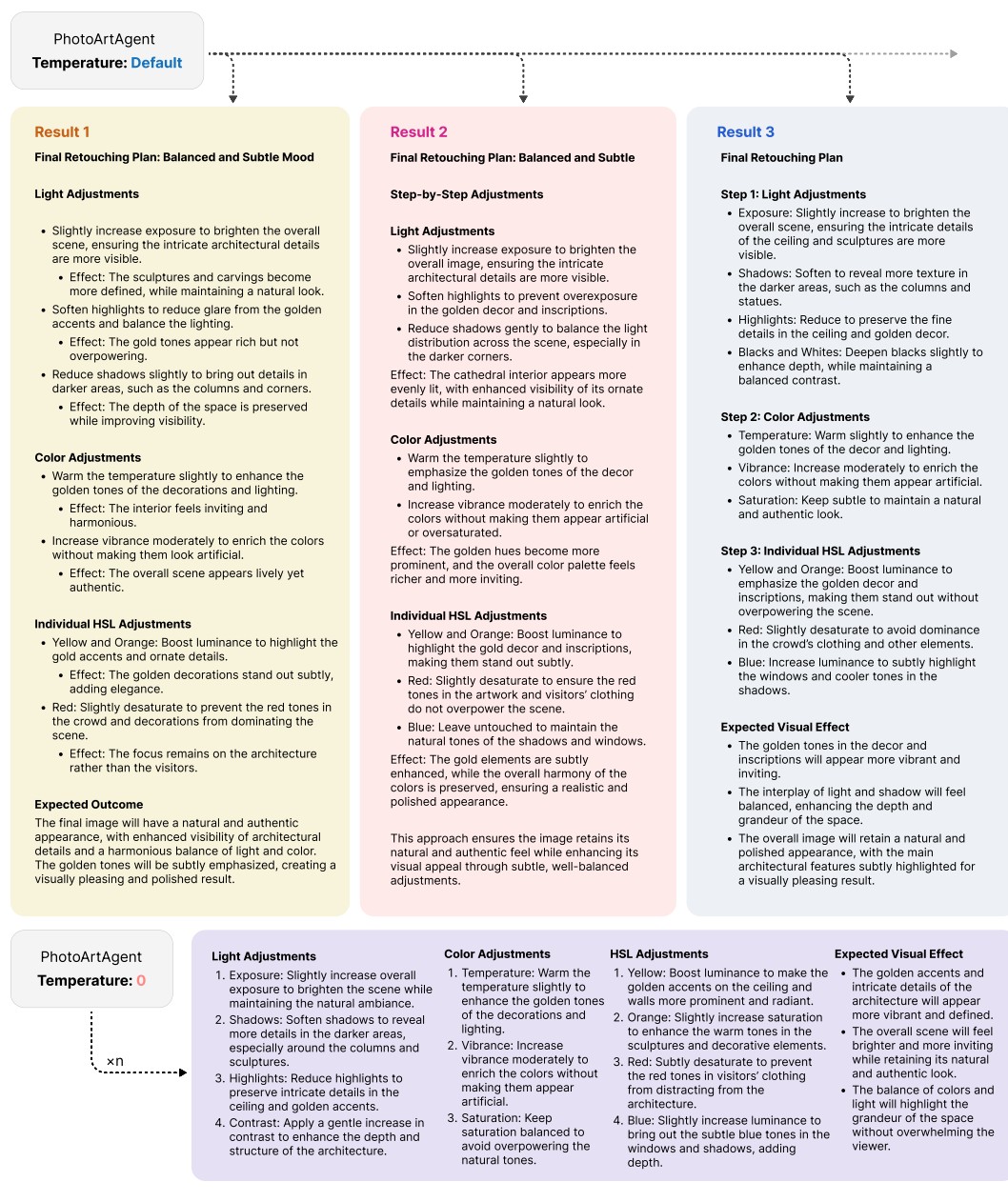

Figure 12: Examples illustrating the impact of temperature on language model outputs. Higher temperatures introduce greater variability and creativity by assigning higher probabilities to less likely tokens, while lower temperatures, such as greedy decoding (temperature = 0), produce deterministic and consistent results.

For the majority of initial parameter selections, our reflection mechanism effectively identifies and corrects errors. Nevertheless, in certain uncommon scenarios, the model remains at risk of producing hallucinations. When this happens, its decisions regarding retouching parameters may become unreliable because it lacks the necessary experience or knowledge for that specific class of images. In such cases, the model might default to a familiar but suboptimal approach.

Figure 11 in our main text illustrates one way to mitigate these issues: by explicitly providing relevant experience and context within the prompt. Our experiments indicate that offering such focused guidance reduces the likelihood of hallucination, improving the reliability of PhotoArtAgent's outputs.

### B.6 FUTURE DIRECTIONS

**More Tools**   Although our current implementation of PhotoArtAgent primarily focuses on Lightroom's light adjustments, color corrections, and Color Mixer HSL manipulations, future work can expand its capabilities to cover the full spectrum of photo-editing tools. For instance, more advanced local adjustments such as spot healing, gradient filters, and AI-based object removal could be integrated. Likewise, third-party applications offering specialized retouching or artistic effects (e.g., skin smoothing, background replacement, or style transfer) could be incorporated into PhotoArtAgent's workflow. By broadening the range of tools accessible to the system, we can further approximate the versatility and creativity of human artists.

**Cooperation with Other Retouching Methods**   Beyond incorporating additional tools, PhotoArtAgent can cooperate with other automated or semi-automated retouching methods to achieve a hybrid approach that combines rule-based strategies, deep learning models, and human-like reasoning. For instance, certain steps – such as noise reduction or super-resolution – could be offloaded to specialized neural networks, with PhotoArtAgent subsequently evaluating and refining the outcome based on its higher-level artistic judgment. Similarly, advanced face- or object-recognition algorithms might handle complex segmentation tasks, while PhotoArtAgent focuses on global or stylistic adjustments. This cooperative framework would not only enhance workflow efficiency but also enable a more holistic retouching pipeline where multiple methods dynamically support each other.

**Knowledge Injection**   One promising direction for mitigating hallucinations and improving the model's domain expertise is the injection of external knowledge. For instance, *Retrieval-Augmented Generation* (RAG) techniques could be employed to equip the language model with references to authoritative databases, user manuals, or style guides. When PhotoArtAgent encounters an unfamiliar scenario – say a specific photography genre or lighting condition – it could query these external sources to gather accurate, context-relevant information before deciding on a course of action. By grounding the model's reasoning in verifiable facts, knowledge injection helps reduce the risk of fabricating parameters or recommendations. Moreover, domain-specific guidance (e.g., portrait retouching best practices, color-matching rules for specific industries, or even historical art references) can be introduced to further refine the creative process. This approach maintains the flexibility and open-endedness of large language models while improving reliability and depth of expertise.

**Assessment Module**   Finally, the design of a dedicated *Assessment Module* could significantly improve the quality and coherence of PhotoArtAgent's retouching outcomes. Such a module might function as an internal critic or feedback loop, evaluating the aesthetic and technical merits of the current edit. For instance, it could assess image sharpness, color balance, exposure consistency, or even subjective measures of visual appeal through learned aesthetic models. The output of this assessment would guide subsequent retouching iterations or even prompt the agent to explore alternative solutions. By formalizing an assessment stage—potentially powered by a second language-based agent, a classical computer vision algorithm, or a hybrid of both – PhotoArtAgent could not only enhance its self – correcting capabilities but also better adapt to different artistic or technical requirements.

## C   MORE EXPERIMENTS DETAILS

### C.1   PROMPT DESIGN

For agent systems built upon pre-trained language models, the design of each step in the *prompt* is as crucial as the broader cognitive workflow. Even the most carefully structured cognitive architecture hinges on clear, detailed prompts to guide the model effectively.

**Character, Background, Ambition, and Task (CBAT).**   Our prompt-design methodology follows a "Character, Background, Ambition, and Task" (CBAT) approach:

- **Character** defines the role and persona of the agent role. By establishing a clear voice and perspective (e.g., an experienced photography tutor or an assistant photo editor), we help the model maintain consistency in style and tone throughout the interaction.

- **Background** provides the context or domain-relevant information needed to reason about the current problem. This may include details on image retouching techniques, the underlying software environment (Lightroom), or user preferences.

- **Ambition** outlines the high-level goals the agent aims to achieve. This ensures the model remains aligned with overall objectives while making step-by-step decisions.

- **Task** specifies the concrete instructions or actions the agent should perform at this stage. For instance, we might instruct the model to "evaluate the current brightness and propose a numerical adjustment," or "suggest possible color grading strategies for a sunset scene."

Each stage of the cognitive architecture is articulated through one or more prompts that follow this CBAT structure. Precisely crafting these prompts helps the agent maintain a coherent workflow and reduces the risk of confusing or contradictory instructions.

**Illustrative Prompts.** In the supplementary materials, we provide examples of how we apply the CBAT method to our PhotoArtAgent framework. It is important to note that there is no single "best" prompt design; our examples merely illustrate one possible approach. In practice, prompt design requires continuous iteration and refinement to address an ever-wider range of images and editing scenarios. By evolving prompts over time – enriching the "Background" with new information, adjusting the "Ambition" to handle stylistic diversity, or sharpening the "Task" instructions – we can steadily improve both the artistic and technical performance of the system.

## C.2 GPT4V-SCORE

We utilized a GPT-based aesthetic evaluation method, with Prompt B.1 (GPT4V-Score) documenting the detailed operations.

## D MORE RESULTS

We provide additional qualitative comparisons in Figure 19 and Figure 20, and we include some complete outputs of our system at the end of the supplementary materials.

**GPT4V-Score prompt:**

**Task Description**
I want you to act as a professional photography expert and color grading specialist. Your task is to critically evaluate the post-processed color grading of a given image based on the following criteria:

**Evaluation Criteria**

- **Aesthetic Quality:** Does the color grading significantly enhance or detract from the visual appeal of the image? Are the colors harmonious, striking, or distractingly poor?
- **Mood and Atmosphere:** Does the color grading effectively convey the intended mood or atmosphere? Does it evoke the intended emotions or feel entirely mismatched?
- **Creativity:** Does the color grading demonstrate originality and creativity? Is it unique and innovative, or does it feel generic and uninspired?

**Scoring Guidelines** Score the image's color grading quality on a scale of 1 to 10, ensuring that all scores (1-10) are used appropriately across evaluations. Avoid clustering scores within a narrow range (e.g., 5-7). Use the following guidelines:

- **1-2:** Extremely poor; the color grading is either chaotic, poorly executed, or completely mismatched to the image. It actively detracts from the image.
- **3-4:** Below average; the color grading feels uninspired, generic, or mildly mismatched. It adds minimal value to the image.
- **5:** Neutral; the color grading is acceptable but unremarkable. It neither enhances nor detracts from the image in any meaningful way.
- **6-7:** Good; the color grading improves the image and fits its mood. It is aesthetically pleasing but lacks strong creativity or originality.
- **8-9:** Excellent; the color grading is creative, impactful, and supports the image exceptionally well. It demonstrates a clear, artistic vision.
- **10:** Masterpiece; the color grading is groundbreaking, highly creative, and flawlessly executed. It transforms the image into a work of art.

**Important Instructions**

- Be highly critical and diverse in your scoring. Ensure that scores from 1 through 10 are all used across evaluations.
- Avoid assigning the majority of scores to the middle range (4-7). Lower scores (1-3) should be used for uninspired or poor work, and higher scores (8-10) should be reserved for exceptional examples.
- Random variation based on quality is acceptable; do not attempt to normalize or cluster scores.
- Output only a single number from 1 to 10, with no explanation, justification, or additional comments.

Figure 13: Prompt for GPT4V-Score evaluation.

**Image retouching content analyzer and approaches thoughts prompt:**

**Character**
You are a professional image retouching artist with expert knowledge in enhancing images by adjusting light and color. Your task is to analyze the content of an image and propose multiple retouching approaches, each highlighting different aspects and evoking distinct moods.

**Background**
This image is part of a creative project where adjustments to light and color are explored to improve its visual appeal and emotional impact.

**Ambition**
Your goal is to propose three distinct retouching approaches for the image. All adjustments must focus on **Light** (exposure, contrast, highlights, shadows, blacks, and whites), **Color** (temperature, tint, vibrance, and saturation), and **individual color tones** (Red, Orange, Yellow, Green, Cyan, Blue, Purple, Magenta). At least one approach must be bold and exaggerated to create a dramatic effect.

**Task**

1. **Image Content Description:** Provide a detailed description of the image's contents, including key objects, subjects, background, and any notable visual elements. Limit the description to 80 words.

2. **General Requirements for Retouching Approaches:**

   - **Light Adjustments:** Describe how exposure, contrast, highlights, shadows, blacks, and whites should be adjusted for each approach. Specify if the scene should be brighter overall, if shadows should be deepened, or if highlights should be softened.
   - **Color Adjustments:** Describe how temperature, tint, vibrance, and saturation should be handled. Indicate whether the image should be warmer or cooler, have richer or more subdued colors, or feature specific hues more prominently.
   - **Individual Color Adjustments:** Specify any adjustments to individual color tones (Red, Orange, Yellow, Green, Cyan, Blue, Purple, Magenta) by describing changes to **Hue**, **Saturation**, and **Luminance** (HSL).
     - Avoid unnecessary color adjustments—only make changes when essential for the desired effect.
     - Typically, adjust no more than two colors unless the user's instructions specify otherwise.
     - Prefer not to adjust Hue unless it is crucial for the intended result.

3. **Retouching Approaches:**

   - **Approach 1:** Provide a detailed retouching style that emphasizes a balanced or subtle mood.
   - **Approach 2:** Provide a retouching style that contrasts with the first, focusing on a different emotional tone or color palette.
   - **Approach 3:** Provide a bold and exaggerated approach designed to create a dramatic or surreal effect.

Ensure that each approach is precise, cohesive, and does not exceed 100 words.

Figure 14: Prompt for Image retouching content analyzer and approaches thoughts prompt.

**Final Retouching Instructions:**

**Character**
You are a professional image retouching artist tasked with finalizing a single retouching approach based on the user's preferences and previous proposals. Your expertise ensures that the final approach integrates key aspects from different suggestions or follows a single selected approach in full.

**Background**
The user has reviewed previous retouching approaches and provided feedback or specific instructions for a final retouching plan that aligns with their creative goals.

**Ambition**
Your goal is to either choose one of the previously proposed approaches that best matches the user's vision or create a new, cohesive retouching approach by combining elements from different suggestions. Ensure that the final approach fully respects the user's instructions and creative intent.

**User Instruction**
User says: *"{user_instruction}"*

**Task**

1. Review the provided retouching approaches and the user's feedback or instructions.
2. Decide whether to:
    - Select a single approach that fits the user's description.
    - Create a new approach that integrates relevant aspects from different suggestions.
3. Final Approach:
    - Describe the adjustments to **Light** (exposure, contrast, highlights, shadows, blacks, and whites) and **Color** (temperature, tint, vibrance, and saturation).
    - For each adjustment, specify **which objects or areas of the image** are most affected and describe the specific details (e.g., "the intricate carvings on the roof are highlighted by a gentle increase in exposure").
    - Explain the expected **visual effect on these objects**, such as "the water reflections appear richer and more defined" or "the sky becomes softer and more inviting."
    - For each **individual HSL adjustment** (Red, Orange, Yellow, Green, Cyan, Blue, Purple, Magenta), explain why it is necessary and describe the expected visual change for specific objects (e.g., "the red tones in the window frames become more vivid to emphasize their ornate design").
    - Organize the description as a step-by-step plan, indicating the sequence of adjustments.

**Guidelines for Description:**

- Avoid providing exact numerical values—focus on explaining how the adjustments affect the image's visual presentation.
- Mention specific objects, areas, and their corresponding changes to help visualize the effect.
- Ensure the approach remains detailed, logical, and cohesive, and does not exceed 100 words.

Figure 15: Prompt for Final Retouching Instructions.

**Histogram Analysis Instructions:**

**Character**
You are a skilled image processing expert specializing in histogram analysis for deriving insights and improvement directions in photo editing. Your expertise lies in identifying key histogram features that are directly relevant to adjustments in **Light** (exposure, contrast, highlights, shadows, blacks, whites) and **Color** (temperature, tint, vibrance, saturation).

**Background**
This histogram represents the tonal distribution of an image, including shadows, midtones, and highlights. Your goal is to interpret the histogram and identify features that guide light and color adjustments to improve the image's visual quality.

**Ambition**
Your objective is to focus on key histogram characteristics—such as spikes, gaps, or shifts—that inform light and color adjustments. These insights will help refine exposure, balance contrast, and enrich color distribution for an optimized artistic effect.

**Task** Analyze the histogram and describe its most important characteristics relevant to **Light** and **Color** adjustments. Highlight any concentration areas (e.g., shadows, midtones, highlights), significant spikes or dips, and tonal imbalances that may require correction. Explain how these features relate to adjustments like exposure correction, shadow lifting, highlight reduction, or vibrance enhancement.
Focus on how these adjustments can address the histogram's distribution and improve the image's tonal balance and color harmony. Your analysis should begin with **"ADJUSTMENT:"** and should be concise and informative, not exceeding 80 words.

Figure 16: Prompt for Histogram Analysis Instructions.

**Parameter Adjustment Instructions:**

**Character**

As a professional image retouching assistant, I am knowledgeable in Lightroom and have extensive experience determining the appropriate parameters based on the user's needs. I understand both the technical aspects of each adjustment tool and their artistic applications in achieving desired visual outcomes.

**Background**

The user is currently retouching the image. The available adjustments are:

- **Light Adjustments:** Exposure (-5.0, 5.0); Contrast (-100, 100); Highlights (-100, 100); Shadows (-100, 100); Whites (-100, 100); Blacks (-100, 100)

- **Color Adjustments:** Temp (2000, 50000); Tint (-150, 150); Vibrance (-100, 100); Saturation (-100, 100)

- **Individual Color Adjustments:** Hue (-100, 100), Saturation (-100, 100), Luminance (-100, 100) for the following color tones: Red, Orange, Yellow, Green, Cyan, Blue, Purple, Magenta.

It is necessary to determine the specific parameters for this function and confirm the purpose of the adjustment. Key considerations include:

- The current state of the image

- The intended purpose of this specific adjustment

- How this adjustment fits into the overall editing workflow

- The technical limitations and optimal ranges of the parameters

- Potential interactions with other adjustments already applied

**Ambition**

The goal is to provide the user with precise parameter suggestions that maximize the visual impact of the image while maintaining natural and professional results.

Through detailed adjustments, help the user achieve their desired image retouching outcome, ensuring that:

- Colors remain natural and well-balanced

- Contrast and brightness enhance image depth without losing detail

- Adjustments complement rather than compete with each other

- The final result maintains professional quality and authenticity

- The editing choices support the image's intended mood and purpose

**Task**

When adjusting **Light settings** (exposure, contrast, highlights, shadows, whites, blacks), you can make larger changes (up to ±70 or more) if they serve the image's artistic intent and improve the visual storytelling. However, if the goal is subtle refinement, small adjustments are also appropriate. Justify your choices based on the visual needs of the scene.

When adjusting **Color settings** (temperature, tint, vibrance, saturation), maintain a balanced approach unless the artistic vision calls for bold color transformations.

For **individual color adjustments** (Red, Orange, Yellow, Green, Cyan, Blue, Purple, Magenta), specify the purpose of adjusting the **Hue**, **Saturation**, or **Luminance**, and describe the expected effect on image elements (e.g., deepening the sky's blue or emphasizing the warmth of a sunset).

Figure 17: Prompt for Parameter Adjustment Instructions.

**Reflection Instructions:**

**Character**

As a professional image retouching evaluator, I have extensive experience in analyzing image adjustments and determining whether the intended goals have been achieved through the applied parameters.

**Background**

The user has applied adjustments to their image based on previously suggested parameters. The adjustment history and the resulting image are provided for evaluation. It is necessary to assess whether the current adjustment has successfully achieved its intended purpose.

**Ambition**

The goal is to provide a thorough and detailed evaluation of the adjustment results and determine whether to proceed to the next adjustment step or refine the current parameters.

Through careful analysis of the resulting images, along with the adjustment history, help guide the user toward optimal image enhancement while maintaining efficiency in the workflow.

You are meticulous in your evaluation and require specific feedback for improvement, especially for basic adjustments like exposure, contrast, highlights, shadows, and whites/blacks.

**Task**

Analyze the adjusted image and determine whether the modification achieved the intended purpose. Pay special attention to foundational **Light** adjustments and evaluate whether they meet the following criteria:

- **Exposure:** Ensure that the overall brightness matches the intended mood and does not wash out or darken key details (e.g., "the exposure was increased to brighten the scene, but important highlights remain controlled").

- **Contrast:** Confirm that contrast levels enhance depth without making midtones muddy or extremes too harsh (e.g., "the contrast improved subject separation without flattening the midtones").

- **Highlights:** Evaluate whether the highlights are appropriately controlled to avoid overexposure in bright areas (e.g., "the highlights in the sky were softened to maintain cloud detail").

- **Shadows:** Assess whether the shadows retain sufficient detail and enhance depth (e.g., "the lifted shadows reveal texture in the foreground without making the image look flat").

- **Whites and Blacks:** Check if the white and black points are balanced to maintain a full tonal range without clipping (e.g., "the whites add a clean brightness, while the blacks anchor the scene without crushing details").

Next, consider whether the **Light** adjustments complement the **Color** adjustments and contribute to an overall cohesive result.

Additionally, evaluate the image for the following:

- **Effectiveness:** Identify specific visual problems that the adjustment aimed to address and state whether these issues were resolved.

- **Balance:** Determine whether the adjustments are consistent with other elements in the image.

- **Side Effects:** Identify any unintended consequences (e.g., "the lifted shadows made the skin tones appear gray").

**Recommendations:**

Based on the evaluation, provide one of the following recommendations:

1. **If the adjustment is successful:**
   - Confirm that the adjustment achieved its purpose and explain why it is satisfactory.
   - Describe any areas that could be fine-tuned for even better results, if necessary.

2. **If the adjustment needs improvement:**
   - Clearly state the issues and suggest corrections (e.g., "reduce contrast slightly or adjust shadow recovery to avoid flattening").
   - Return `False` when calling the function `satisfactory`.

Your evaluation must be highly detailed and actionable, focusing on foundational adjustments and highlighting any discrepancies that could affect the visual quality. Provide clear, constructive feedback for refining the adjustments and achieving the intended visual goals.

Figure 18: Prompt for Reflection.

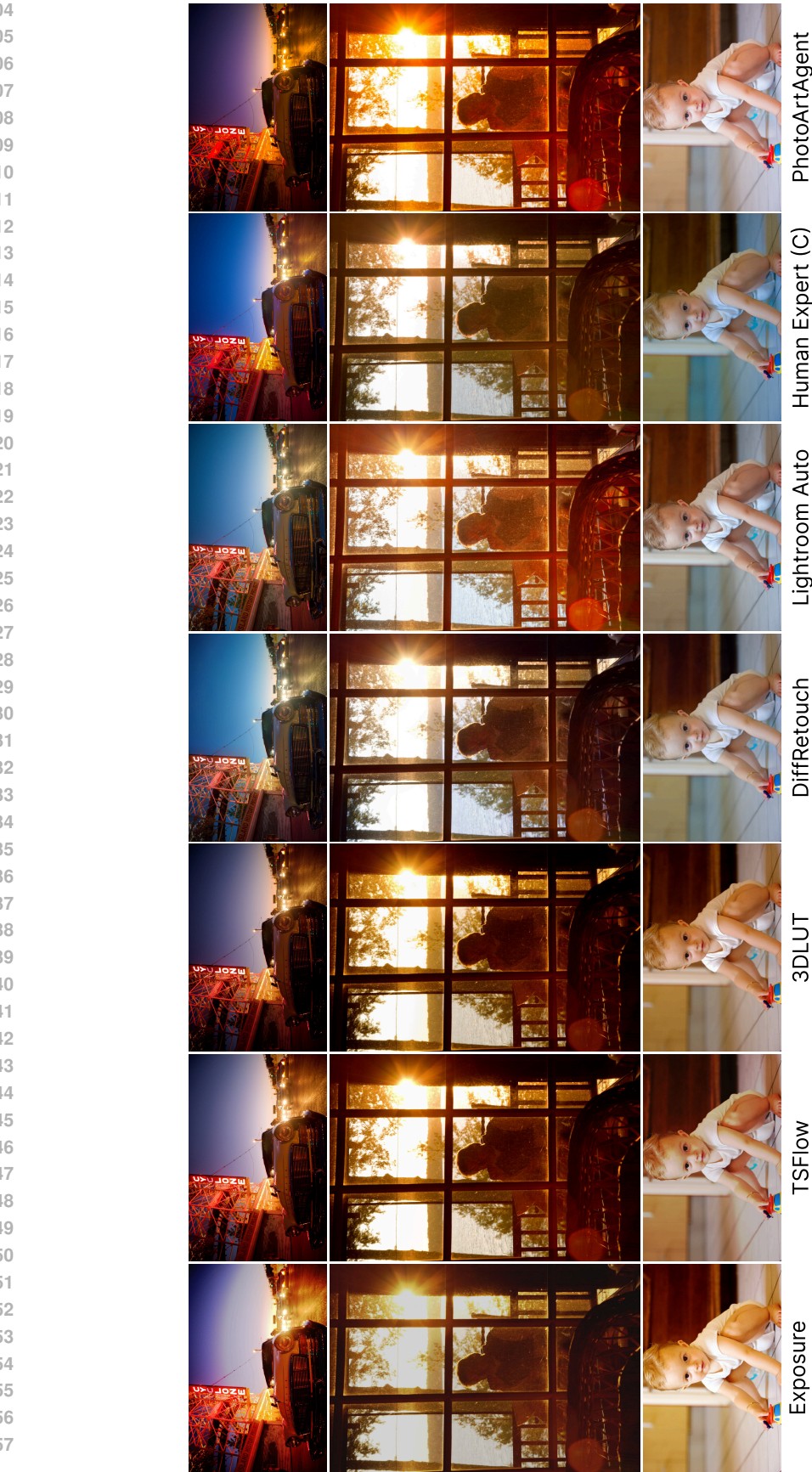

Figure 19: Visual comparison of retouching results across different methods.

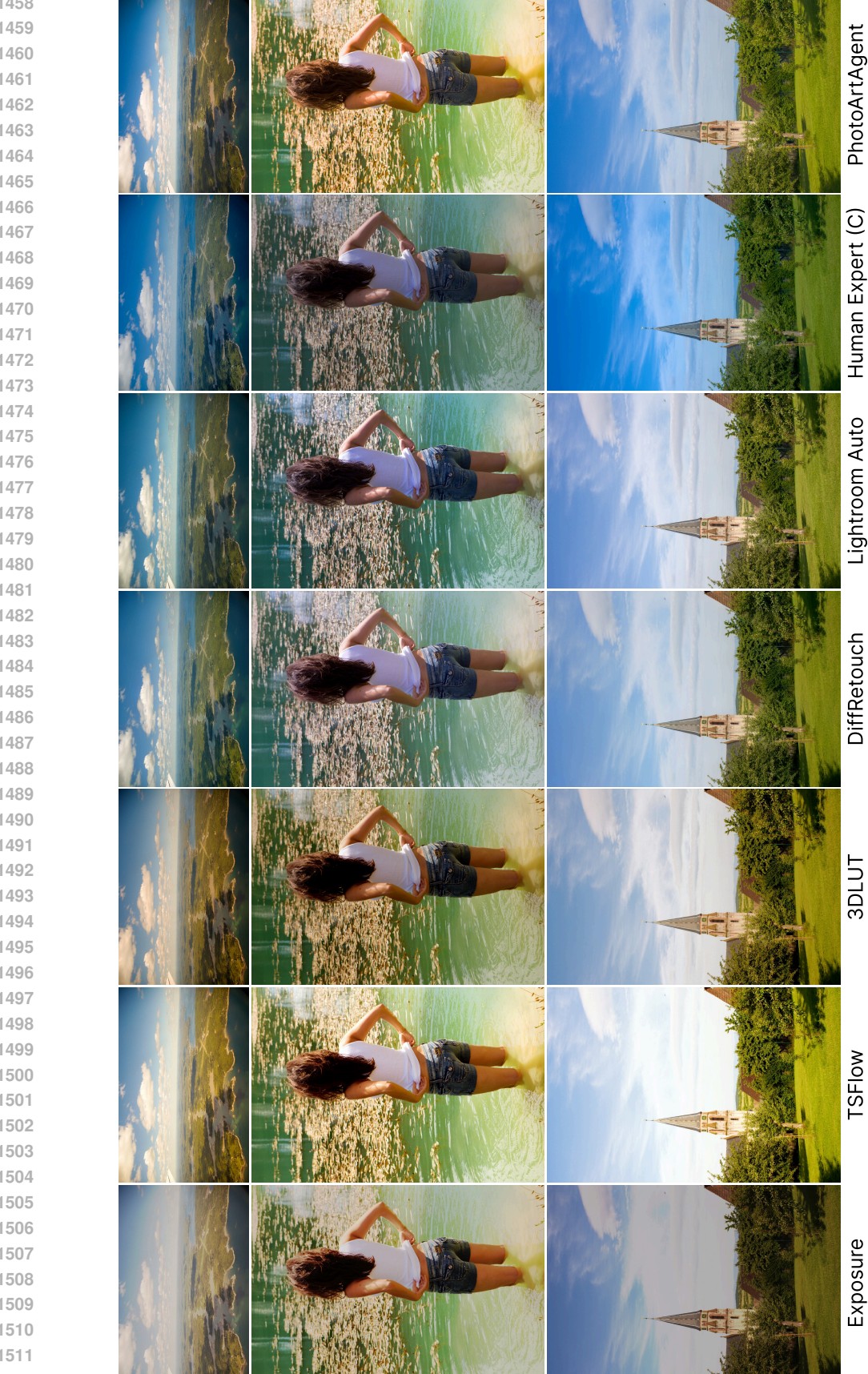

Figure 20: Visual comparison of retouching results across different methods.

### D.1 More Results

## Example 1

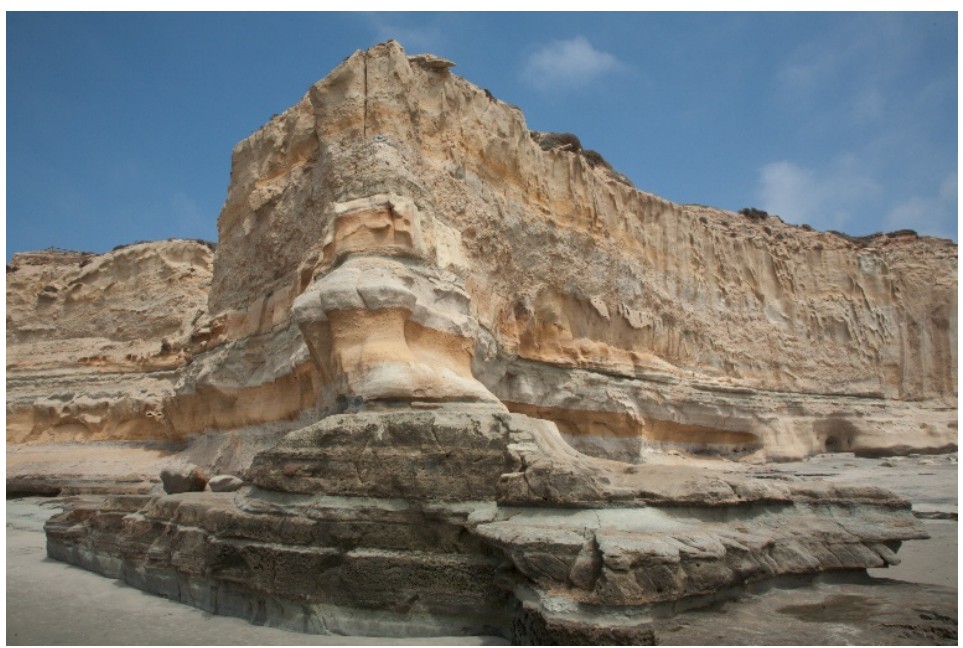

User input image

## Image Content Description

The image portrays an impressive coastal rock formation, featuring beige
and brown layered cliffs eroded by natural processes. The foreground
showcases horizontal rock strata, and the background reveals more rugged
cliffs under a clear blue sky with sparse clouds. The scene has a dry
and arid feel, emphasizing the geological textures and colors of the nat-
ural terrain.

## Retouching Approaches

### Approach 1: Balanced Natural Enhancement

- **Light Adjustments:** Slightly increase exposure and contrast for clarity,
  gently lift shadows, and reduce highlights to preserve sky details.
- **Color Adjustments:** Add subtle warmth to the temperature and slightly
  boost vibrance to enhance natural tones.
- **Individual Color Adjustments:** Slightly increase luminance for yellows
  and oranges to emphasize the rock textures and warmth. Minor desatura-
  tion of blues for a more neutral sky tone.

### Approach 2: Cool and Muted Atmosphere

- **Light Adjustments:** Reduce exposure and deepen shadows for a moodier
  appearance. Lower whites and highlights to tone down brightness.
- **Color Adjustments:** Cool the temperature for a slightly bluish cast,
  reduce vibrance, and tone down saturation for a subdued aesthetic.

- **Individual Color Adjustments:** Desaturate yellows and oranges slightly for a less warm look, and enhance cyan and blue luminance for a gentle atmospheric effect.

### Approach 3:  Bold and Dramatic Effect

- **Light Adjustments:** Dramatically boost contrast, deepen shadows, and increase whites for a stark, high-contrast effect.
- **Color Adjustments:** Intensify warmth with stronger saturation of earthy tones while keeping the sky vibrantly blue.
- **Individual Color Adjustments:** Amplify saturation and luminance of yellows and oranges to make the cliffs stand out prominently. Slightly darken blues for a deep and striking sky.

Each approach provides a distinct interpretation of the scene, catering to different aesthetic and emotional preferences.

## Final Retouching Approach: Balanced Natural Enhancement

### Step 1:  Light Adjustments

- **Exposure:** Slightly brighten the image to bring out details in the cliff and foreground.
- **Contrast:** Increase contrast to emphasize the rock layers and textures.
- **Shadows:** Gently lift shadows to reveal subtle details in the darker rock strata in the foreground.
- **Highlights:** Reduce highlights slightly to retain detail in the sky and softened cloud patterns.

### Step 2:  Color Adjustments

- **Temperature:** Warm the scene minimally to enhance the earthy tones of the rock formation.
- **Vibrance:** Add a touch of vibrance to enrich the sandstone hues without overpowering the natural look.

### Step 3:  Individual Color Adjustments (HSL)

- **Yellow and Orange:** Slightly increase luminance to highlight the sandstone's layered texture, making it stand out more under soft lighting.
- **Blue:** Decrease saturation marginally to prevent the sky from appearing too saturated, ensuring it remains a supportive background.

### Expected Result

- The cliffs gain a warm and inviting glow, with details in the layered structure becoming more pronounced.
- The sky takes on a soft, subdued appearance, drawing focus to the natural textures of the rocky landscape.
- Overall, the image feels natural and authentic, creating a visually appealing and harmonious composition.

## Histogram Analysis (iteration 1)

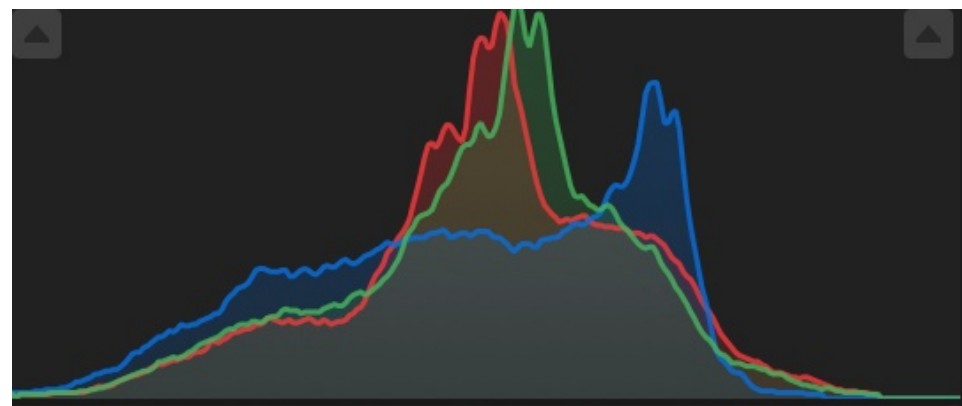

Histogram of the input image

The provided histogram indicates a well-distributed tonal range primarily centered in the midtones, with balanced representation of shadow, midtone, and highlight information. The overall tonal range does not suggest significant underexposure or overexposure, as there are no prominent spikes at either extreme of the histogram. The image appears to have a good black point, with the left edge of the histogram tapering off, implying the presence of true blacks but no heavy clipping in the shadows. Likewise, the white point on the far right edge is similarly defined, suggesting that the image avoids highlight clipping and retains detail in bright areas, though increasing brightness might be an option without risking detail loss.

The midtones exhibit a concentrated and broad peak, particularly in the green and red channels, indicating sufficient contrast and depth in this tonal range. However, the blue channel shows a slightly wider distribution across the tonal range, which may result in cooler tones in parts of the image. This variation among RGB channels implies a slight imbalance in white balance, with the potential for the image to lean towards cooler or greenish hues. Red and green peaks dominate, suggesting warmer areas within the image, while the blue's spread could lead to subtler, cooler elements, affecting overall harmony.

No Sharp or abnormal spikes were observed, meaning the image likely lacks major processing artifacts or harsh transitions. Overall, the histogram shows an image with a naturally balanced tonal range, offering room for slight fine-tuning in both light and color adjustments to achieve the intended aesthetic effect.

## Image Adjustment Parameters (iteration 1)

### Basic Adjustments

| Parameter | Value |
|---|---|
| Exposure | +0.5 |
| Contrast | +20 |
| Highlights | −20 |
| Shadows | +20 |
| Whites | +10 |
| Blacks | −10 |
| Temp (Temperature) | 5800 |
| Tint | +5 |
| Vibrance | +25 |
| Saturation | +15 |

**Color Adjustments (HSL)**

| Color | Hue | Saturation | Luminance |
|---|---|---|---|
| Orange | 0 | +10 | +10 |
| Yellow | 0 | +10 | +15 |
| Cyan | 0 | −10 | +5 |
| Blue | 0 | −10 | −10 |

The adjustments aim to bring out the rock formation's warm tones while maintaining the natural blue of the sky. Brightness and contrast adjustments enhance detail and depth, and selective color adjustments improve structure in key elements without over-modifying the scene's natural look.

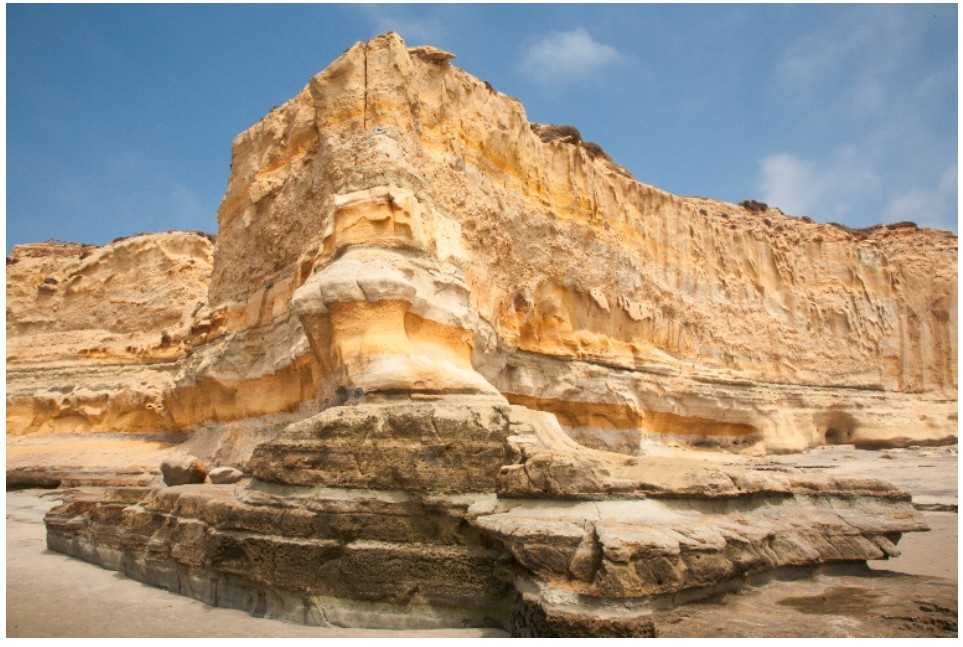

Retouched image (iteration 1)

## Reflection (iteration 1)

**The adjustment is NOT satisfactory.** The adjustments did not fully align with the global style requirements. The highlights remain slightly too

strong, causing minor distraction in the rock details. Shadows could be further lifted to clarify darker parts of the image, and white balance adjustment is required to reduce the cooler undertones in select areas of the rocks.

## Histogram Analysis (iteration 2)

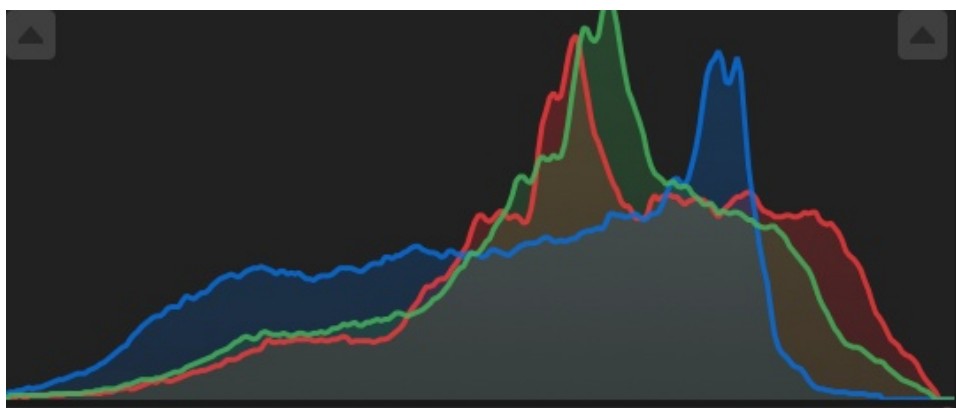

Histogram of the retouched image (iteration 1)

The histogram represents a well-balanced tonal range, with a smooth distribution of shadows, midtones, and highlights. There is no significant concentration at either end, suggesting that the image avoids both underexposure and overexposure. The black point on the far-left edge is defined without clipping, implying the presence of natural shadows without losing detail in the darkest areas. Similarly, the white point on the far-right edge is adequately represented, avoiding any extreme highlight clipping and maintaining detail in bright parts of the image. The central peaks in the red and green channels indicate warm midtones typical of earthy or organic tones, while the wider distribution of the blue channel suggests cooler elements in some parts of the image, potentially lending subtle complexity to the overall color tone. However, the imbalance among channels, with red and green dominating over blue, could slightly shift the white balance towards a warmer or greenish tint. No unusual spikes or gaps are notable, indicating the image's tonal transitions are smooth and free from artifacts. Overall, the histogram suggests an image with a natural tonal range, ready for minor refinements in color balance and contrast to enhance harmony and clarity.

## Image Adjustment Parameters (iteration 2)

### Basic Adjustments

| Parameter | Value |
|---|---|
| Exposure | +0.5 → +0.3 |
| Contrast | +20 → +25 |
| Highlights | −20 → −30 |
| Shadows | +20 → +25 |
| Whites | +10 → +15 |
| Blacks | −10 |
| Temp (Temperature) | 5800 |
| Tint | +5 → +8 |
| Vibrance | +25 → +20 |
| Saturation | +15 → +10 |

**Color Adjustments (HSL)**

| Color | Hue | Saturation | Luminance |
|---|---|---|---|
| Orange | 0 | +10 → +15 | +10 |
| Yellow | 0 | +10 → +15 | +15 → +20 |
| Cyan | 0 | −10 | +5 |
| Blue | 0 | −10 → −15 | −10 |

The adjustments aim to emphasize the warmth and intricacy of the rock formation while preserving the natural tone of the sky. Shadows and whites are adjusted to reveal details in darker and brighter areas, and color calibrations enhance the rock and sky contrast harmoniously.

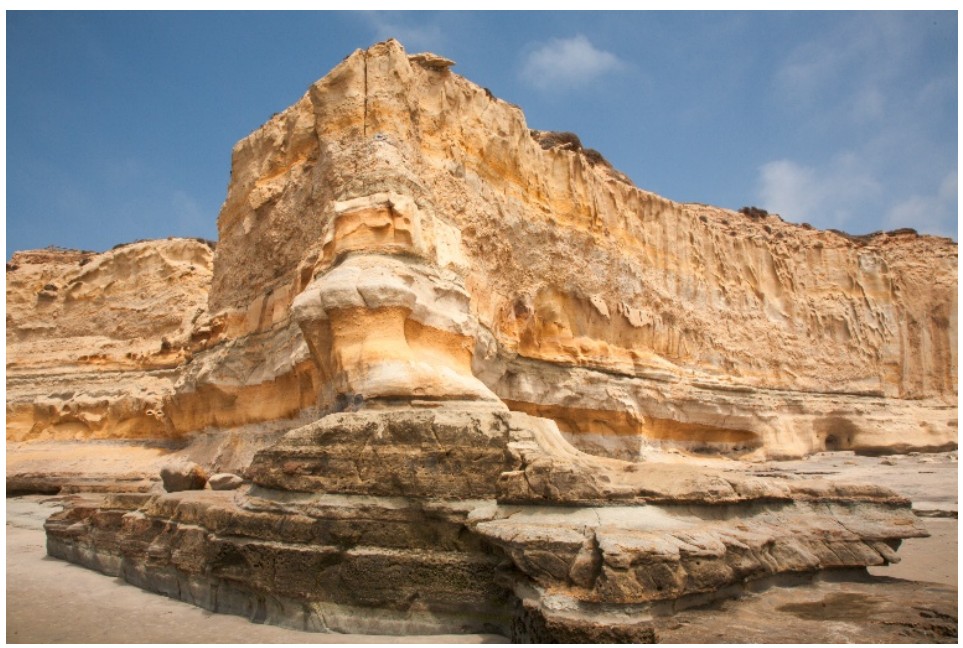

Retouched image (iteration 2)

## Reflection (iteration 2)

**The adjustment is satisfactory.** The adjustments successfully align with the user's requirements, enhancing the warmth and intricate details of

the rock formation while maintaining a natural and harmonious look across the entire image.  Balance between the tones and colors was achieved, especially in the shadow and sky context.

## Example 2

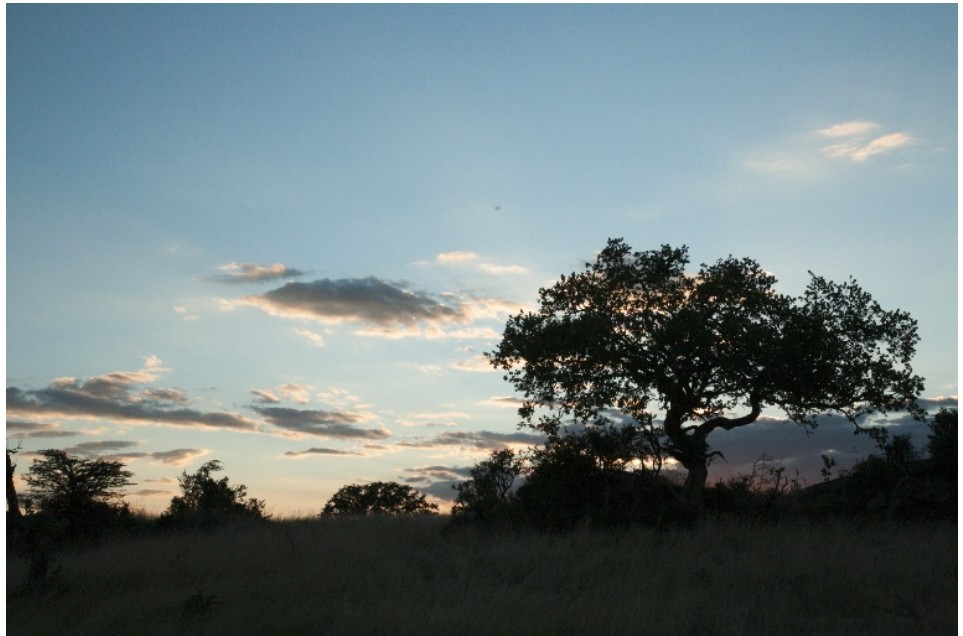

User input image

### Image Content Description

The image showcases a serene outdoor landscape at dawn or dusk. A prominent tree dominates the right foreground, with its dark silhouette contrasting against a subtle gradient of a pastel-colored sky. The background features scattered smaller trees and bushes, with soft clouds illuminated by warm sunlight. The foreground contains a grassy field in shadow, while the horizon line fades gently into the scene.

### Retouching Approaches

#### Approach 1: Balanced and Natural Mood

- **Light:** Slightly brighten the shadows while maintaining the highlights for a balanced dynamic range. Deepen blacks slightly to preserve the tree silhouette.
- **Color:** Increase vibrance subtly to enhance sky tones without oversaturation. Maintain a neutral-to-warm white balance.
- **Individual Colors:** Enhance Yellow and Orange luminance to emphasize the clouds and the sunlit area. Slightly boost the Blue saturation to enrich the sky.

#### Approach 2: Warm and Inviting Mood

- **Light:** Brighten exposure and soften shadows to evoke a softer, golden-hour effect. Reduce highlights slightly to avoid glare.
- **Color:** Warm the temperature by increasing yellow and red tones. Slightly boost saturation for richer, warm hues.
- **Individual Colors:** Increase Orange and Red saturation and luminance for a glowing sunset effect. Soften Blue tones by reducing their saturation.

### Approach 3:  Bold and Dramatic Mood

- **Light:**  Intensify contrast by deepening shadows and brightening highlights.  Slightly underexpose the image to accentuate the silhouette effect.

- **Color:**  Cool the temperature moderately to create a stark contrast between the sky and tree.  Increase vibrance dramatically.

- **Individual Colors:**  Boost Blue and Cyan luminance for a vivid sky. Deepen Orange and Yellow tones to create a strong focal point on the clouds.  Shift Hue of Blue slightly towards Cyan for an ethereal feel.

## Final Retouching Approach

The image will benefit from a retouching approach combining subtle mood enhancement from approaches 1 and 2.  Minor adjustments to light and color will be implemented to balance natural authenticity and visual appeal effectively.

### Light Adjustments

- **Increase overall exposure moderately** (e.g., +1.5 or more) to enhance visibility in the dark foreground, drawing out subtle details in the grass.

- **Decrease highlights and whites significantly** to balance bright areas and eliminate glare in the sky.

- **Boost shadows and blacks slightly** to reveal texture in darker areas while maintaining the silhouette of the tree.

- **Enhance contrast** to deepen the visual separation of foreground and background elements.

### Color Adjustments

- **Warm the temperature marginally** to emphasize a golden, soft atmospheric tone without drifting into artificial warmth.

- **Increase vibrance slightly** to enrich the pastel hues of the sky while avoiding dramatic oversaturation.

- **Target Orange and Yellow luminance enhancements** to highlight the glowing edges of the clouds, providing a striking focal point.

- **Boost Blue saturation gently** to accentuate the depth of the sky.

### Expected Visual Outcome

- The tree silhouette will retain its dramatic and natural look.

- The sky will become more visually appealing with interplay between warm and cool tones.

- The grassy foreground will gain subtle detail and texture, unifying the composition while keeping it understated.

## Histogram Analysis (iteration 1)

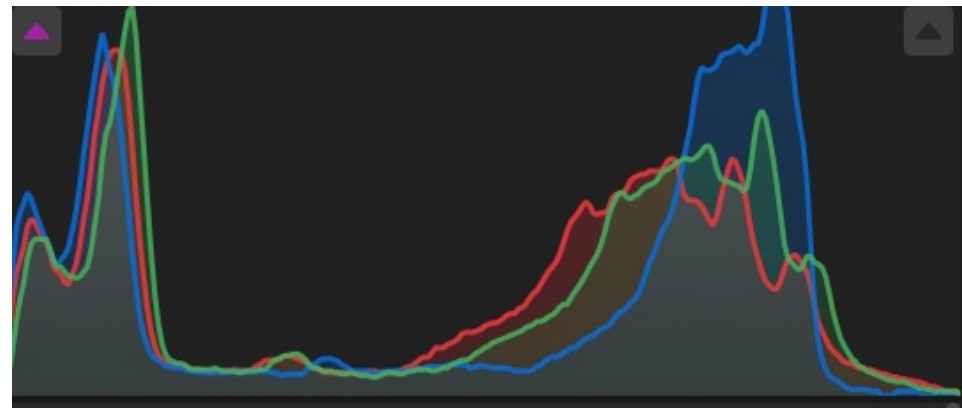

Histogram of the input image

The histogram displays a dual-peak distribution with distinct data clus-
tering at both shadows (left side) and highlights (right side), suggest-
ing a contrast-heavy image.  There is a strong concentration of blue in
the highlights, as indicated by the high peak of the blue channel on the
far right, likely contributing a cool tone to the brighter areas.  Con-
versely, the red and green channels demonstrate a relatively balanced
distribution, albeit with lower peaks, which may temper the overall vi-
brancy.  The black point appears well-defined with the RGB channels ex-
tending fully to the far-left edge, though the elevated peaks in shadows
indicate the image may contain crushed blacks, losing some shadow de-
tails.  Regarding the white point, the histogram reveals a slight domi-
nance of blue in the brightest areas, and little to no clipping is evi-
dent, maintaining detail in highlights.  This creates a clean tonal range
but with a potential cool color cast.  Midtones are underrepresented,
suggesting reduced tonal depth overall, resulting in a potentially high-
contrast image with limited gradation between shadow and highlight areas.
For white balance, the dominance of blue, particularly in highlights,
signals a cooler image that may lack warmth.  Overall, the histogram sug-
gests that contrast adjustments to balance shadow and highlight details
and fine-tuning of color temperature to neutralize the cool cast would
improve tonal and color harmony.

## Image Adjustment Parameters (Iteration 1)

### Basic Adjustments

| Parameter | Value |
|---|---|
| Exposure | +1.5 |
| Contrast | +20 |
| Highlights | −60 |
| Shadows | +30 |
| Whites | −35 |
| Blacks | +15 |
| Temp (Temperature) | 5500 |
| Tint | 0 |
| Vibrance | +15 |
| Saturation | +10 |

### Color Adjustments (HSL)

| Color | Hue | Saturation | Luminance |
|-------|-----|------------|-----------|
| **Orange** | 0 | 10 | 15 |
| **Yellow** | 0 | 10 | 20 |
| **Blue** | 0 | 15 | 10 |

These adjustments brighten the image while preserving contrast and detail. Warming the tones slightly enhances the golden-hour effect, and individual color tweaks improve the clouds' glow and sky depth.

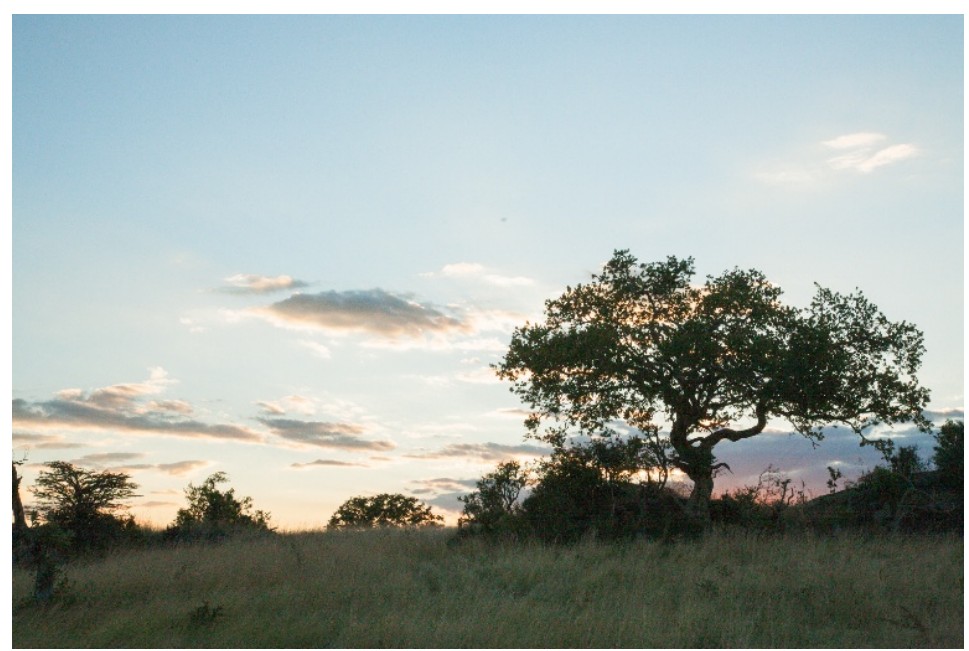

Retouched image (iteration 1)

## Reflection (iteration 1)

**The adjustment is NOT satisfactory.** The current adjustments do not fully satisfy the user's requirements. The image retains a cool tone that conflicts with the warm, visually appealing look intended by the user. Highlights are not sufficiently reduced, which risks minor overexposure in the brightest areas of the sky. Additionally, shadow details remain underrepresented due to crushed blacks shown in the histogram. Midtone adjustments are also insufficient, causing limited tonal depth and overall balance. Suggest warming the temperature further, enhancing vibrance, reducing the highlights by -85, and brightening shadows more to unlock details.

## Histogram Analysis (iteration 2)

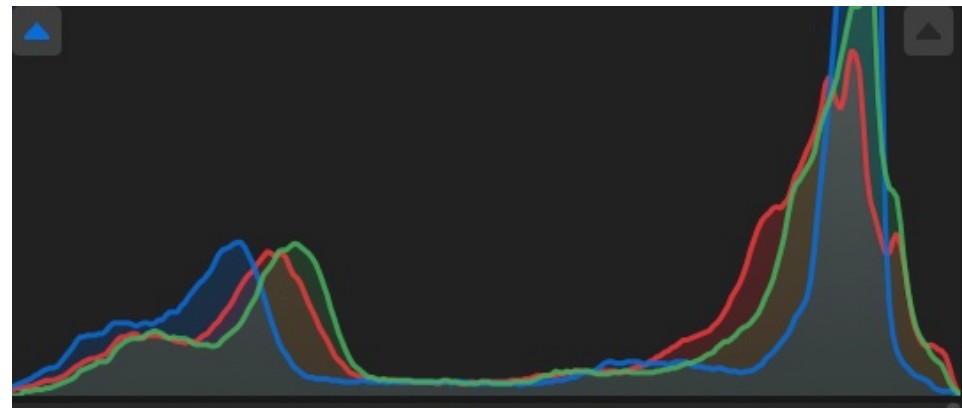

Histogram of the retouched image (iteration 1)

The histogram reveals a heavy concentration of tones at both ends of
the spectrum, indicating a high-contrast image with a potential lack of
midtone details. The black point on the far-left edge is well-defined,
showing all three RGB channels reaching the axis, but the elevated clus-
tering in the shadows may suggest crushed blacks, leading to a loss of
shadow detail. On the right side, the histogram highlights a strong
presence of highlights without evident clipping, signifying intact bright
details, though the dominance of the blue channel in this region creates
a noticeable cool tone. The underrepresentation of midtones results in
an image with stark dynamics but insufficient tonal gradation. The red
and green channels are comparably balanced across the tonal range, pre-
venting dramatic color imbalances, but the blue channel's prominence,
particularly in highlights, yields a cooler overall hue. For optimal im-
provements, the tonal range should be broadened with greater emphasis on
midtones, shadows can be lifted to recover texture, and color tempera-
ture adjustments can balance the cool tone for a more natural or visually
coherent effect.

## Image Adjustment Parameters (Iteration 2)

### Basic Adjustments

| Parameter | Value |
|---|---|
| Exposure | +1.5 |
| Contrast | +20 |
| Highlights | −60 → −75 |
| Shadows | +30 → +50 |
| Whites | −35 → −40 |
| Blacks | +15 → +10 |
| Temp (Temperature) | 5500 → 6500 |
| Tint | 0 |
| Vibrance | +15 → +20 |
| Saturation | +10 |

### Color Adjustments (HSL)

| Color | Hue | Saturation | Luminance |
|-------|-----|------------|-----------|
| Orange | 0 | +10 → +15 | +15 → +20 |
| Yellow | 0 | +10 | +20 → +25 |
| Blue | 0 | +15 → +10 | +10 → +15 |

The parameters are chosen to balance the visible tones while softening the current harsh contrasts. Light adjustments recover shadow detail, minimize highlights, and preserve the silhouetted tree while improving tonal depth. Color modifications enhance sky vibrancy and warmth while retaining its natural appearance and emphasizing subtle orange and yellow cloud details.

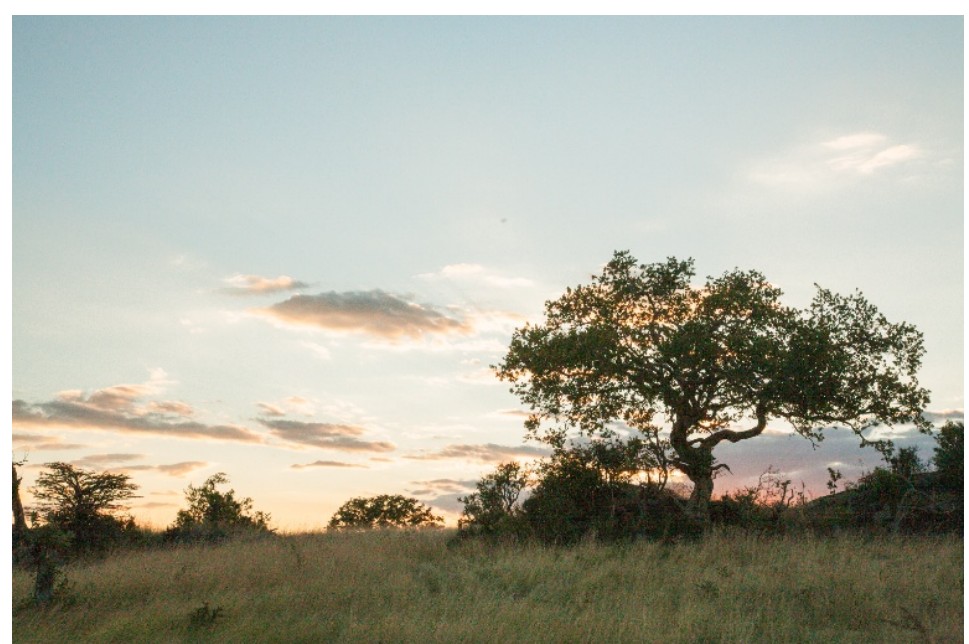

Retouched image (iteration 2)

## Reflection (iteration 2)

**The adjustment is NOT satisfactory.** The current adjustment does not meet the user requirements for a natural-looking and balanced outcome. The overall exposure remains somewhat low, shadow details are still obscured, and tonal balance lacks smooth gradation, as evidenced by the histogram's steep contrast profile. Additionally, the cool tone from blue-dominant highlights conflicts with the user's request for a subtle, natural, and slightly warmer appearance.

## Histogram Analysis (iteration 3)

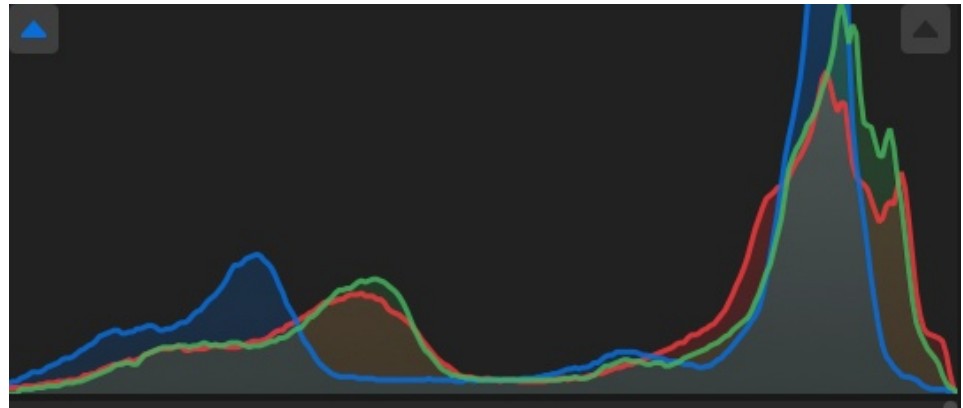

Histogram of the retouched image (iteration 2)

The histogram illustrates a dual-peak tonal distribution with clusters in both shadow and highlight regions. Shadows are well-defined at the far left, with some risk of crushed blacks due to sharp peaks. Highlights dominate on the far right, especially in the blue channel, which indicates a cool bias in brighter image areas. The midtones appear under-represented, suggesting reduced tonal depth and a heavy contrast profile. The RGB dominance of blue in the highlights suggests a cool temperature, with red and green balancing the lower tonal areas. Adjustments should focus on recovering midtones, reducing shadow crushing, and mitigating the dominance of cool highlights by slightly warming the color balance for tonal and color harmony.

## Image Adjustment Parameters (Iteration 3)

### Basic Adjustments

| Parameter | Value |
|---|---|
| Exposure | +1.5 |
| Contrast | +20 |
| Highlights | −75 → −70 |
| Shadows | +50 → +30 |
| Whites | −40 → −50 |
| Blacks | +10 → +20 |
| Temp (Temperature) | 6500 → 5500 |
| Tint | 0 → +10 |
| Vibrance | +20 → +15 |
| Saturation | +10 |

### Color Adjustments (HSL)

| Color | Hue | Saturation | Luminance |
|---|---|---|---|
| Orange | 0 | +15 → +10 | +20 → +15 |
| Yellow | 0 | +10 | +25 → +20 |
| Blue | 0 | +10 | +15 |

The adjustments focus on balancing shadows and highlights to improve visibility while retaining tonal depth, slightly warming the temperature for natural appeal, and enriching the vibrance of the sky to emphasize the

interplay of warm and cool tones. The specific orange, yellow, and blue adjustments highlight the luminous clouds and sky depth.

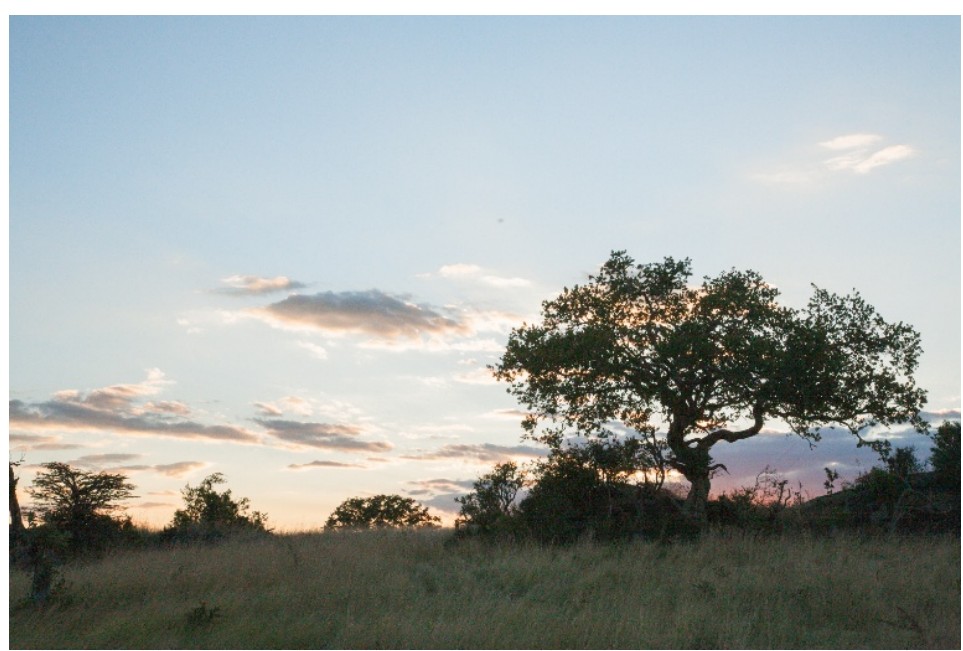

Retouched image (iteration 3)

## Reflection (iteration 3)

**The adjustment is satisfactory.** The adjustments successfully balanced shadows and highlights, maintaining the natural silhouette effect while emphasizing subtle details in the brighter areas. The color temperature was correctly warmed moderately, blending harmoniously with the sky's gradients and enhancing visual appeal while preserving authenticity.

**Example 3**

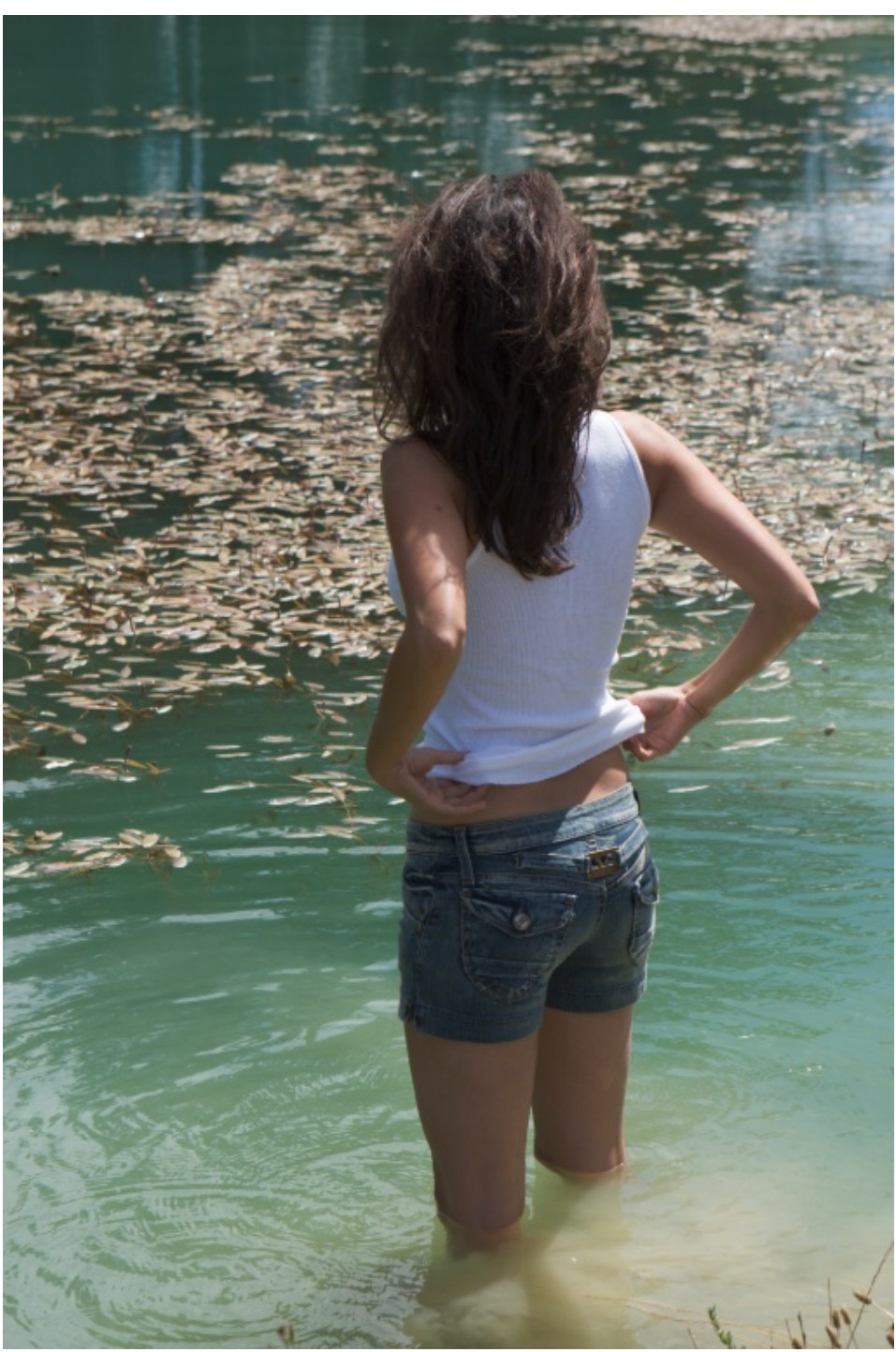

User input image

**Image Content Description**

This image shows a woman with back-facing posture, standing in shallow, greenish-blue water. She is wearing a white tank top and denim shorts,

with her hands resting on her lower back and hip. The water surface is filled with floating leaves. The scene is outdoors in natural light, with a tranquil and casual atmosphere.

## Retouching Approaches

### Approach 1: Subtle and Natural Look

- **Light Adjustments:** Slightly increase exposure and reduce highlights for a softer look. Boost shadows to enhance details in the darker areas of her clothing. Balance whites while increasing contrast moderately.
- **Color Adjustments:** Adjust temperature to retain a neutral tone, slightly enhancing vibrance for natural hues. Keep saturation balanced.
- **Individual Color Adjustments:** Enhance Green's saturation and luminance to make the water more vibrant. Slightly increase Blue luminance for clarity.

### Approach 2: Warm and Dreamy Tone

- **Light Adjustments:** Increase exposure for a warmer, airy scene. Reduce contrast and deepen blacks for a softer focus.
- **Color Adjustments:** Warm up the temperature and increase vibrance to make the colors pop. Slightly enhance overall saturation.
- **Individual Color Adjustments:** Shift Green hue towards Yellow for a golden effect. Boost Orange and Yellow luminance for a sunny ambiance.

### Approach 3: Bold and Dramatic Impact

- **Light Adjustments:** Increase contrast heavily and reduce shadows for high drama. Focus on brightening highlights for sharp contrasts.
- **Color Adjustments:** Cool down the temperature for a surreal feel, significantly increasing vibrance and saturation.
- **Individual Color Adjustments:** Shift Green hue towards Cyan and boost its saturation dramatically. Deepen the blues in shadows and increase Blue saturation and luminance for a vivid, otherworldly effect.

## Final Retouching Plan

### Light Adjustments

- **Exposure:** Slightly lower the exposure overall to maintain a slightly darker and subtler tone while preserving highlights for a natural glow. The white tank top and edges of the subject become more defined without overpowering the environment.
- **Contrast:** Increase slightly to add depth, especially enhancing the separation between the subject and the water backdrop. The denim shorts appear more textured, and water ripples gain definition.
- **Shadows:** Lift the shadows modestly in the subject's hair and darker regions of the clothing. Provides slight detail visibility in shadow areas, ensuring realism.

### Color Adjustments

- **Temperature & Tint:** Keep the color temperature neutral to maintain authenticity while slightly enhancing the vibrance. Increase the vibrance slightly to ensure the water and leaves look more striking yet natural. Overall colors stay understated, with just enough enhancement to highlight textures.

### Individual HSL Adjustments

- **Green:** Boost saturation and luminance. Makes the water appear fresh and subtly vibrant without feeling artificial.
- **Blue:** Slightly increase luminance for gentler tones in water reflections. Smoothens and highlights water surfaces.
- **Yellow/Orange:** Subtly enhance luminance to brighten leaves floating on the surface. Creates more dimensionality and clarity for the foliage.

### Final Visual Outcome

The image retains its authentic, natural allure with slightly more vivid water and enhanced subject detail. The balance of warmer tones in the leaves and cooler water creates a visually harmonious and understated effect.

## Histogram Analysis (iteration 1)

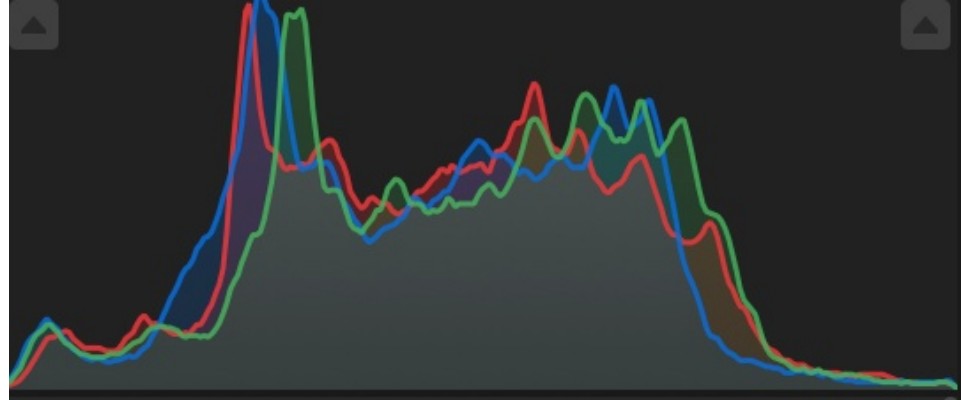

Histogram of the input image

The provided histogram signifies a generally balanced distribution of tonal values, though a few key characteristics emerge. The tonal range stretches across most of the histogram but lacks pronounced extremities. Data clustering on the left edge shows a defined black point with limited shadow clipping, indicating that shadow detail is still preserved. Conversely, the right side of the histogram lacks significant data, revealing an insufficient white point and brightness levels, making the image potentially dull or lacking highlight detail. Midtone data is moderately distributed, with some peaks signifying areas of focused tonal compression, potentially contributing to a perception of limited contrast or tonal depth.

In terms of color balance, the histogram shows discrepancies among the RGB channels. The green channel is more pronounced in the mid-to-shadow range, suggesting a greenish tint overall. The red channel, peaking slightly more in the midtones, hints at a subtle warm hue, while the blue channel remains lower, likely causing a cooler shadow tone. These deviations imply a color imbalance, leading to an image skewed towards cooler greens with reduced warmth or vibrancy in highlights.

Lastly, the histogram lacks extreme spikes or abrupt gaps, showing an overall smooth distribution. The absence of these features suggests minimal processing artifacts but also indicates that the image may lack dynamic tonal variation. Taken together, the histogram reflects subdued

contrast and color imbalances that could affect the image's depth and overall harmony.

## Image Adjustment Parameters (Iteration 1)

### Basic Adjustments

| Parameter | Value |
|---|---|
| Exposure | +0.2 |
| Contrast | +15 |
| Highlights | −20 |
| Shadows | +15 |
| Whites | −10 |
| Blacks | +10 |
| Temp (Temperature) | 5700 |
| Tint | +5 |
| Vibrance | +20 |
| Saturation | +10 |

### Color Adjustments (HSL)

| Color | Hue | Saturation | Luminance |
|---|---|---|---|
| Orange | 0 | +10 | +5 |
| Yellow | 0 | +10 | +10 |
| Green | 0 | +20 | +15 |
| Blue | 0 | +10 | +10 |

This adjustment enhances the overall mood by subtly enriching the natural colors, minimizing harsh highlights, and balancing shadows for detail. The slightly warmer temperature and vibrance emphasize the greenery and water tones while retaining a calm, authentic look.

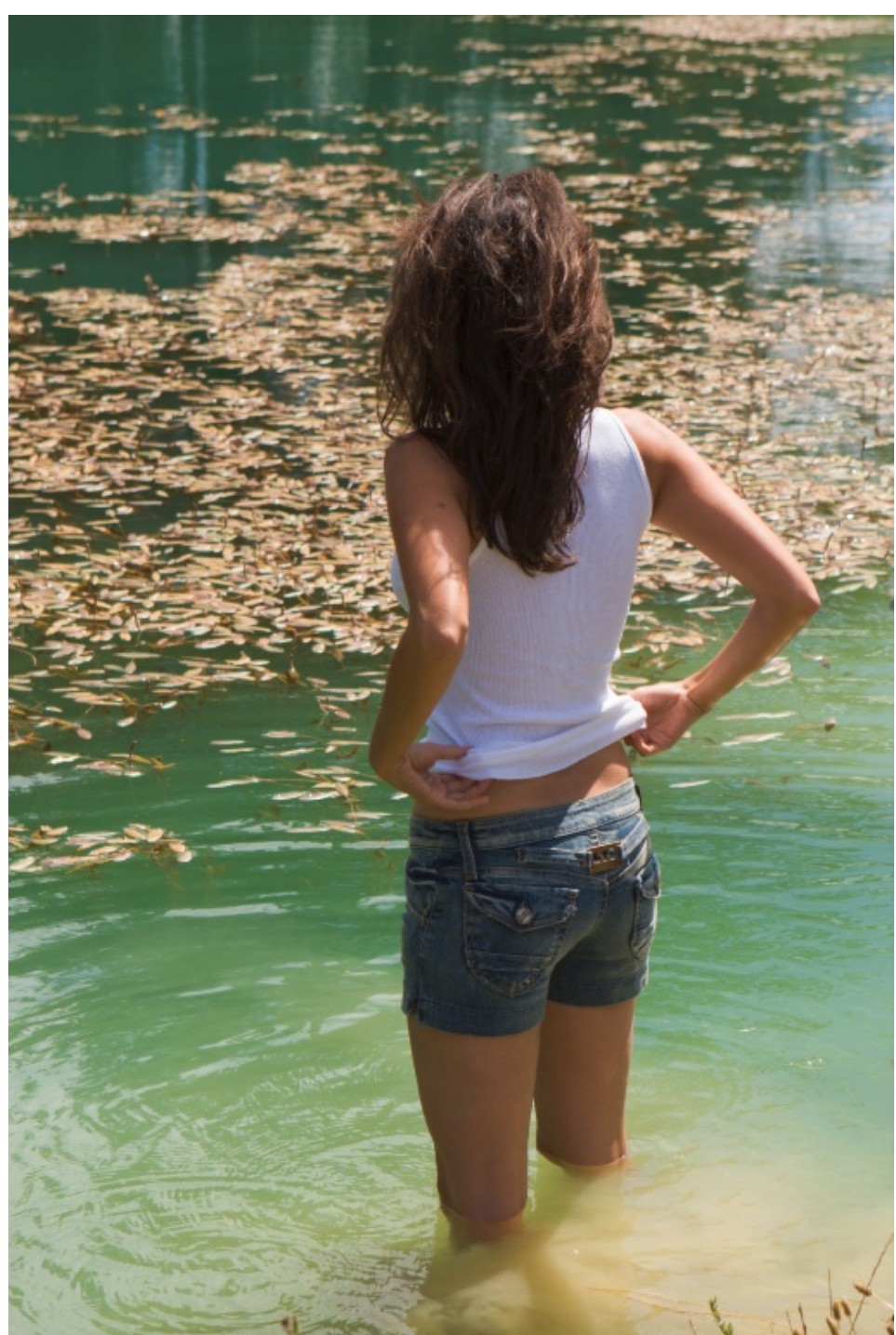

Retouched image (iteration 1)

## Reflection (iteration 1)

**The adjustment is NOT satisfactory.** The current retouching needs improvement due to imbalanced whites and highlights, which cause a slight lack of visual clarity. Adjust whites and increase contrast slightly for improved subject focus and depth, while ensuring natural color harmony is retained.

## Histogram Analysis (iteration 2)

Histogram of the retouched image (iteration 2)

The histogram shows a generally balanced tonal range, with a few areas requiring attention. The exposure is neither overly high nor low but is moderately placed, suggesting the image has more of a standard lighting balance. The black point is well-defined on the far-left edge, with some data presence, meaning that the shadows are preserved without significant clipping. However, the white point on the far-right edge lacks substantial data, indicating the image could benefit from improved brightness or enhanced highlights to add some vibrancy and depth.

In the midtones, significant clustering is observed, pointing to a compressed tonal range that might contribute to a slightly flat appearance. This indicates insufficient tonal separation and limited contrast, reducing the image's sense of depth and liveliness. As for color balance, the green channel dominates in certain regions, particularly in the midtones, which can give the image a cooler or greenish cast. The red channel offers some warmth but appears less pronounced, whereas the blue channel has moderate peaks and remains neutral in most tonal regions. The color imbalance, with the green channel leading, could affect the overall harmony and visual appeal of the image by leaning it towards cooler tones.

Lastly, the lack of extreme spikes or hard gaps suggests that the tonal range transitions smoothly, indicating no significant processing artifacts. Overall, the histogram hints at a need to brighten the highlights, increase contrast in the midtones, and carefully adjust the color balance to infuse energy and correct tonal flatness for enhanced depth and harmony.

## Image Adjustment Parameters (Iteration 2)

### Basic Adjustments

| Parameter | Value |
|---|---|
| Exposure | +0.2 → +0.3 |
| Contrast | +15 → +25 |
| Highlights | −20 → −25 |
| Shadows | +15 → +20 |
| Whites | −10 → −5 |
| Blacks | +10 → +15 |
| Temp (Temperature) | 5700 → 6000 |
| Tint | +5 → +10 |
| Vibrance | +20 → +30 |
| Saturation | +10 → +15 |

**Color Adjustments (HSL)**

| Color | Hue | Saturation | Luminance |
|---|---|---|---|
| Orange | 0 | +10 → +15 | +5 → +10 |
| Yellow | 0 | +10 → +15 | +10 |
| Green | 0 | +20 → +25 | +15 → +20 |
| Blue | 0 | +10 → +20 | +10 → +15 |

The adjustments aim to energize the image by enhancing depth and bringing out vibrant tones in the water and greenery. Increasing brightness, contrast, and shadows adds dimensionality, while the warmer temperature improves harmony and sunlight-like warmth. This provides a natural yet striking outcome suitable for professional quality.

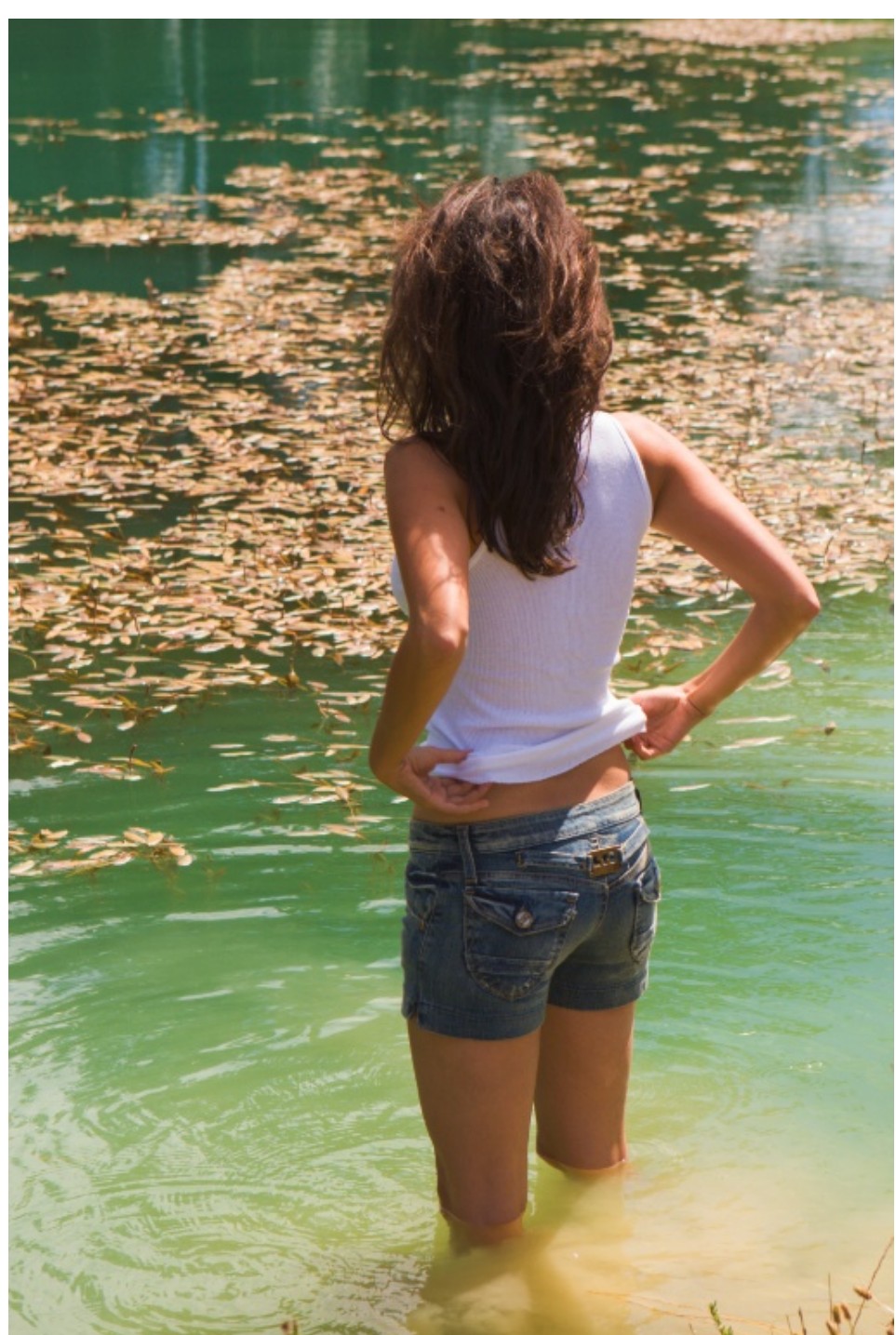

Retouched image (iteration 2)

## Reflection (iteration 2)

**The adjustment is satisfactory.** The left image has been effectively enhanced in the right image by improving contrast, vibrancy, and natural exposure levels. The adjustments achieve user requirements by maintain-

ing realism while enhancing fine details, creating a balanced natural yet pleasing rendering.

## Example 4

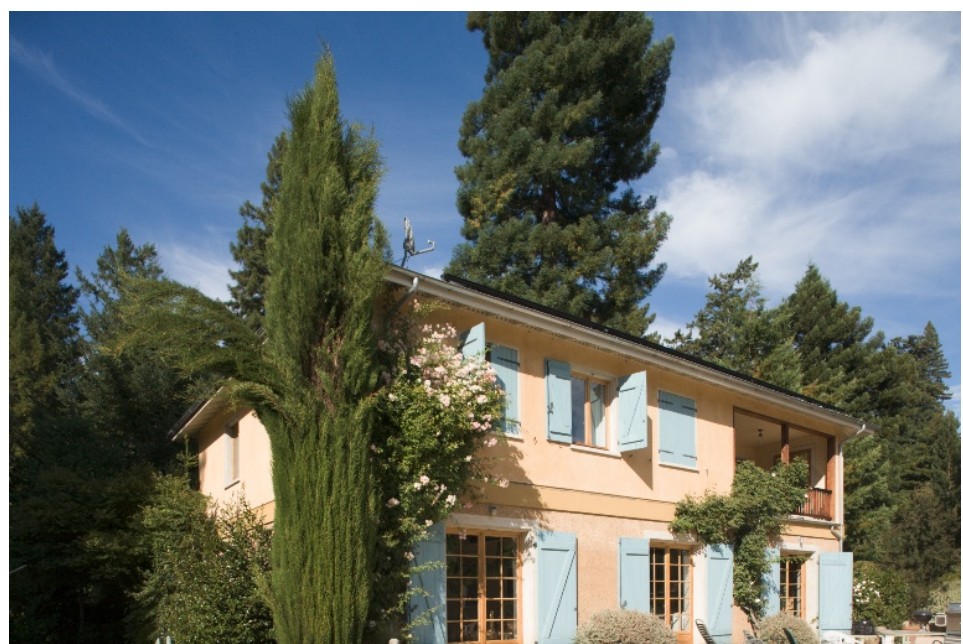

User input image

## Image Content Description

This image depicts a sunny, serene scene featuring a charming house with soft yellow walls and pastel blue shutters. It is surrounded by lush greenery, including a tall cypress tree and colorful flowering bushes. In the background, large, dense trees border the property, while the bright blue sky with wispy clouds enhances the tranquil atmosphere.

## Retouching Approaches

### Approach 1: Balanced and Natural Mood

- **Light Adjustments:** Increase overall brightness slightly and enhance contrast for clarity, softening shadows slightly to retain texture in greenery.
- **Color Adjustments:** Slightly increase vibrance and warmth to emphasize the sunny environment while maintaining natural tones.
- **Individual Color Adjustments:** Boost yellows and greens in luminance and saturation for a lively feel without oversaturation. Bring out the light blue of the shutters gently.

### Approach 2: Cool and Calm Mood

- **Light Adjustments:** Lower overall exposure slightly, decreasing highlights for a subdued feel, while deepening shadows for added depth.
- **Color Adjustments:** Cool the color temperature to evoke a relaxed ambiance and reduce vibrance for a slightly muted effect.
- **Individual Color Adjustments:** Desaturate greens and yellows lightly to balance the temperature, while enhancing the blue shutters' saturation and luminance for contrast.

### Approach 3: Dramatic and Surreal Effect

- **Light Adjustments:** Increase contrast dramatically, deepen blacks, and brighten whites for a bold composition. Slightly add vignetting.
- **Color Adjustments:** Intensify vibrance and saturation across warm tones (yellows, oranges) for a striking, surreal summer aesthetic.
- **Individual Color Adjustments:** Shift yellow hues slightly toward orange, increase saturation of greens for an almost other-worldly lushness, and emphasize the blues in the sky and shutters for a dreamlike clarity.

## Final Retouching Plan

### Light Adjustments

- Slightly boost overall exposure, maintaining a touch of darkness for understated appeal.
- Increase contrast moderately to bring out details in the house facade and greenery.
- Adjust shadows and highlights to preserve textures in the trees and structure, ensuring the scene remains realistic.

### Color Adjustments

- Warm up the color temperature slightly to emphasize the sunny setting.
- Increase vibrance gently, just enough to make colors pop without feeling artificial.

### HSL Adjustments

- **Yellow:** Enhance saturation and luminance to make the house's walls appear slightly more vibrant.
- **Blue:** Brighten and enhance the shutters and sky for a cheerful yet soft appearance.
- **Green:** Boost saturation slightly, emphasizing the lush greenery without oversaturating it.

### Expected Outcome

The image will maintain its authentic and polished look, with slightly enriched colors and refined contrasts that subtly highlight the harmony between the house and its natural surroundings.

## Histogram Analysis (iteration 1)

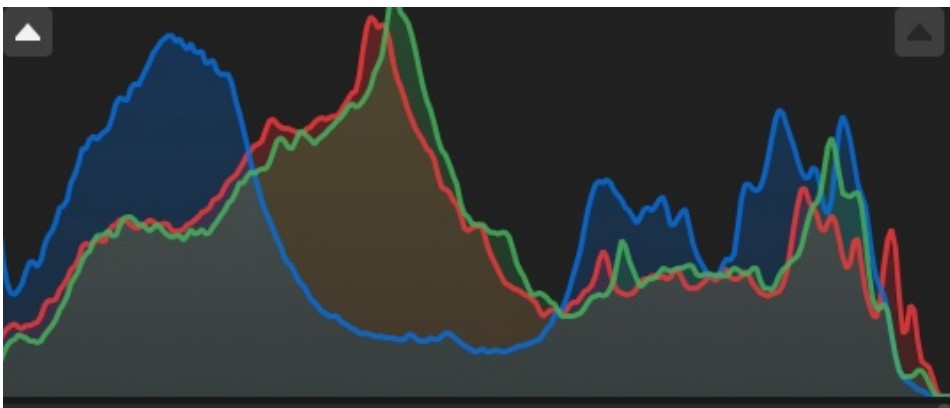

Histogram of the input image

The histogram presents a relatively even tonal range, slightly favoring midtones and highlights. The distribution shows some peaking toward the brighter and midtone areas (center-right), while shadows (far-left) and blacks are represented lightly, suggesting that the image may lack strong dark tones. This absence of a prominent black point could result in a subdued sense of depth. Conversely, the white point appears mostly intact but without significant data clustering at the far-right edge, indicating that the image isn't overexposed but might benefit from slightly enhanced brightness in highlights. The color balance across the RGB channels suggests a mild imbalance, with the green and red channels somewhat pronounced in the midtones and highlights. This could imply a slightly warm and yellowish cast in the image. The blue channel is well-represented but less prominent compared to the green and red, which aligns with a warmer tone overall. Other notable features include some uneven peaks toward the mid and brighter tones in the green and red channels, which might indicate an overemphasis on certain hues of greens and yellows. These could benefit from subtle adjustments to restore harmonious saturation and to prevent saturation clumping. Overall, the histogram reveals an image that leans slightly toward a soft, warm aesthetic but lacks strong tonal depth in the shadows and blacks, limiting dynamic range and contrast.

## Image Adjustment Parameters (Iteration 1)

### Basic Adjustments

| Parameter | Value |
| --- | --- |
| Exposure | +0.3 |
| Contrast | +15 |
| Highlights | −10 |
| Shadows | +10 |
| Whites | +20 |
| Blacks | −10 |
| Temp (Temperature) | 5500 |
| Tint | 0 |
| Vibrance | +10 |
| Saturation | +5 |

### Color Adjustments (HSL)

| Color | Hue | Saturation | Luminance |
| --- | --- | --- | --- |
| Yellow | 0 | +10 | +15 |
| Green | 0 | +5 | +5 |
| Blue | 0 | +10 | +15 |

The adjustments aim to maintain the natural and polished look of the image while slightly enriching its vibrancy and depth. The exposure and contrast adjustments enhance clarity, while color temperature helps emphasize the sunny ambiance. Individual color tuning highlights the yellow house walls, lush greenery, and blue sky and shutters without oversaturating.

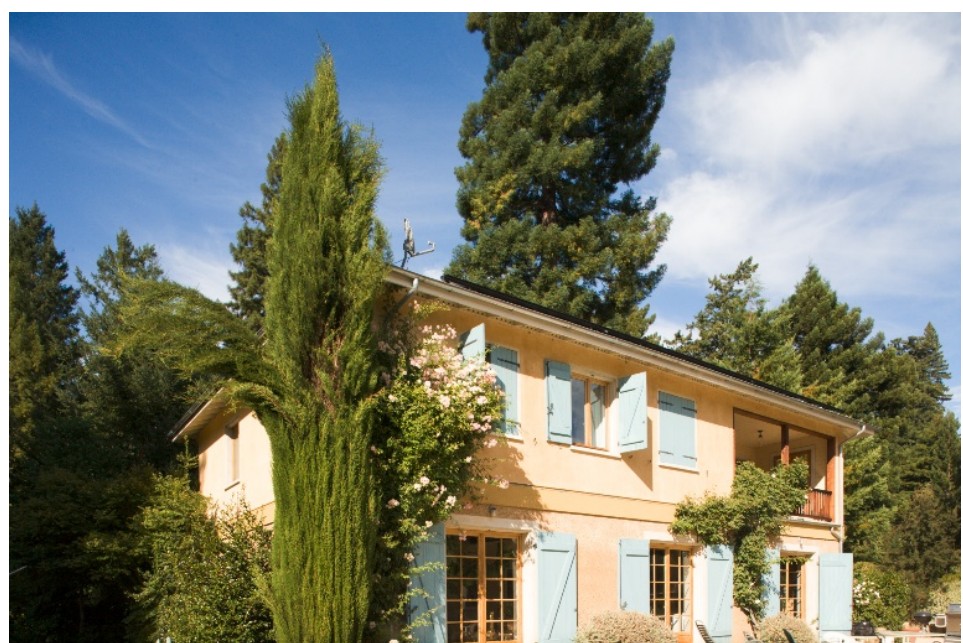

Retouched image (iteration 1)

## Reflection (iteration 1)

**The adjustment is NOT satisfactory.** The original adjustment plan ex-
hibits a warm and pleasant mood; however, the current implementation
does not align fully with user requirements. The image lacks sufficient
shadow depth as indicated by the histogram, resulting in reduced over-
all contrast and a flat appearance. The blue channel is subdued compared
to the reds and greens, leaving the image slightly warmer than intended
without achieving harmony. Further balancing of colors, an emphasis on
enhancing tonal depth, and counterbalancing the yellowish cast in some
hues are needed to better meet user expectations. Exposure can remain
stable, but highlights could use a slight boost for brilliance.

## Histogram Analysis (iteration 2)

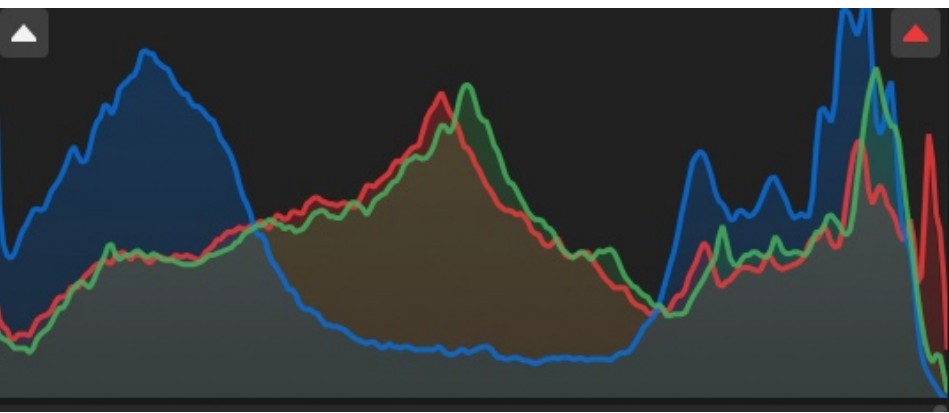

Histogram of the retouched image (iteration 1)

The histogram reveals several key aspects of this image's tonal and color balance. First, there is a moderate-to-high clustering of data in the midtones and highlights, with distinct peaks in the red and green channels, indicating a warm-toned image. The image likely reflects a sunny, well-lit scene dominated by yellows, greens, and possibly muted reds, as these are suggested by the dominant RGB data. However, the blue channel appears underrepresented in both the shadows and highlights, skewing the temperature toward warmth and possibly creating a yellowish cast.

The far-left region of the histogram, representing shadows and blacks, shows a limited presence of deep tones, suggesting the black point is not well-defined. This indicates a lack of rich, deep shadows and could cause the image to feel flat or lack contrast in darker regions. On the other hand, the far-right edge displays adequate data, with no significant highlight clipping, meaning that while highlights are accounted for, they are just at the threshold of sufficient brightness.

Overall, the histogram highlights the following issues: a warm bias in color temperature caused primarily by stronger red and green channels with diminished blue presence, and a constrained tonal range in the shadows. These factors result in an image that could benefit from enhanced shadow depth and possible cooling of the overall tone for better tonal balance and warmth control. This would lead to improved overall contrast, depth, and color harmony in the scene.

## Image Adjustment Parameters (Iteration 2)

### Basic Adjustments

| Parameter | Value |
|---|---|
| Exposure | +0.3 |
| Contrast | +15 → +25 |
| Highlights | −10 |
| Shadows | +10 → +15 |
| Whites | +20 → +10 |
| Blacks | −10 → −20 |
| Temp (Temperature) | 5500 → 5000 |
| Tint | 0 |
| Vibrance | +10 → +15 |
| Saturation | +5 |

### Color Adjustments (HSL)

| Color | Hue | Saturation | Luminance |
|---|---|---|---|
| Orange | 0 | 0 → +10 | 0 → +5 |
| Yellow | 0 | +10 → +20 | +15 |
| Green | 0 | +5 → +10 | +5 → +10 |
| Blue | 0 | +10 → +15 | +15 |

The adjustments aim to enhance the natural beauty of the image by slightly increasing exposure and contrast to add depth. Shadows and blacks deepen for more contrast, while highlights reduce slight brightness peaking. Warmth and vibrance highlight the sunny look, and specific color adjustments improve the house's walls, greenery, and sky without oversaturation.

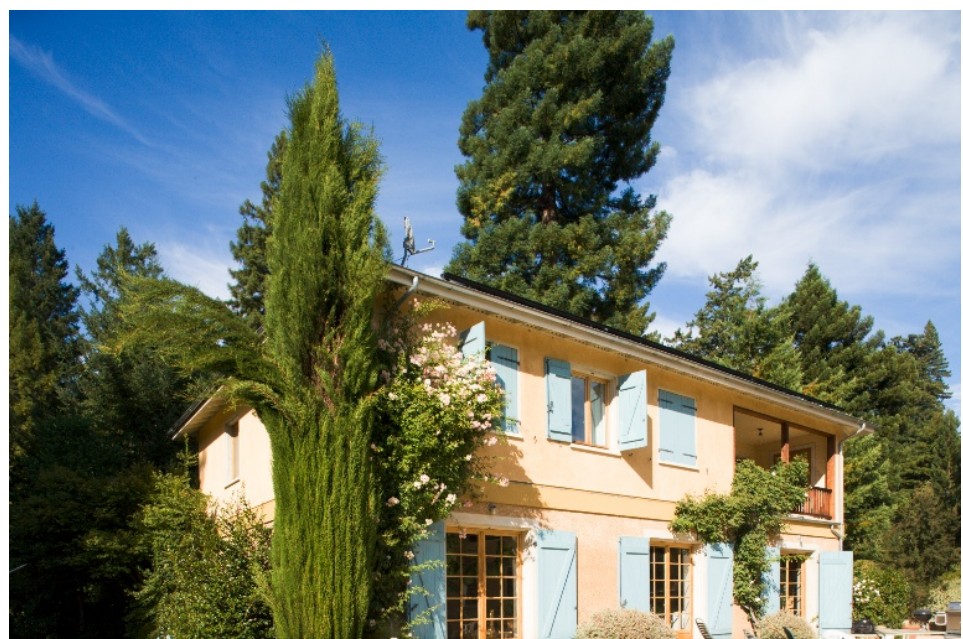

Retouched image (iteration 2)

## Reflection (iteration 2)

**The adjustment is satisfactory.** The adjustments successfully enhance the natural appeal of the image, maintaining a realistic tonal balance with a warm yet not overpowering temperature. Shadows are deepened for better dimension, while specific color adjustments bring out the house, greenery, and sky without feeling artificial.

**Example 5**

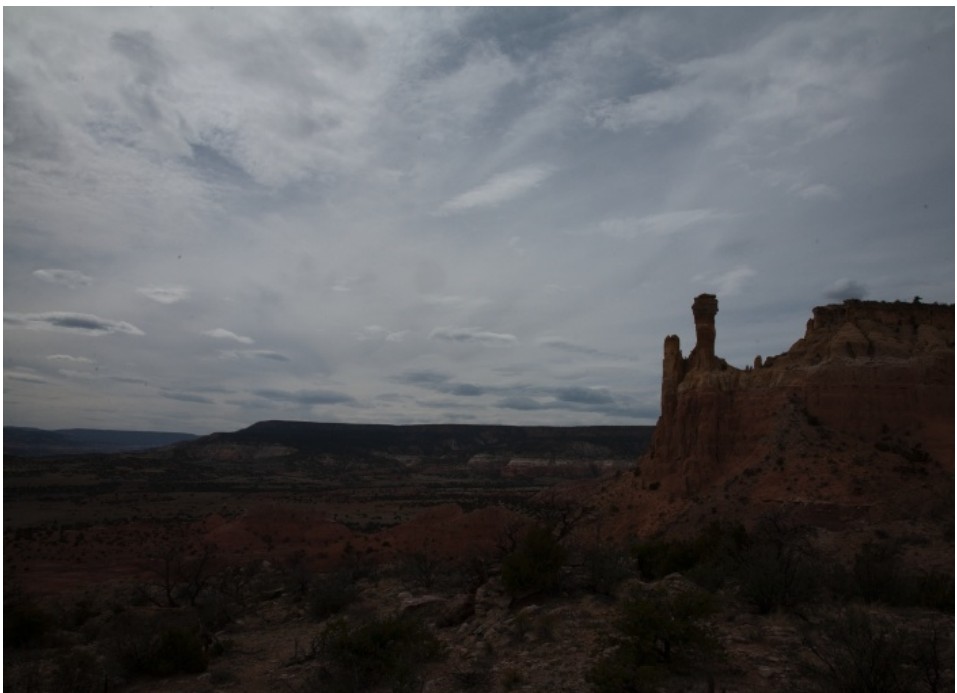

User input image

## Image Content Description

The image showcases a rugged desert landscape featuring a prominent rock
formation resembling a solitary spire or pillar on the right. The ter-
rain consists of reddish-brown sand and sparse vegetation, with a back-
drop of expansive sky filled with thick clouds. The dark and moody
lighting creates a calm yet dramatic atmosphere.

## Retouching Approaches

### Approach 1: Balanced and Subtle Mood

- **Light Adjustments:** Slightly brighten the image by increasing exposure
  and highlights while softening shadows to reveal more detail. Add a
  moderate contrast boost for balance.
- **Color Adjustments:** Warm the temperature slightly to bring out earthy
  tones. Enhance vibrance gently while keeping saturation natural.
- **Individual Color Adjustments:** Increase the luminance of reds and yel-
  lows slightly to emphasize the rock formations. Subtly boost the satu-
  ration of blues to enhance the sky.

### Approach 2: Cool and Serene Desert Scene

- **Light Adjustments:** Focus on a cooler, softer scene by lowering high-
  lights and increasing exposure slightly. Retain some shadow depth for
  texture.
- **Color Adjustments:** Shift the temperature cooler and lower vibrance
  slightly for a subdued aesthetic.

- **Individual Color Adjustments:** Muted reds and yellows for less stark rock colors, while blues and cyans are brightened to create an airy, serene look in the sky.

### Approach 3:  Bold and Dramatic Effect

- **Light Adjustments:** Deepen shadows and increase contrast heavily while slightly brightening highlights for an intense, dramatic feel.
- **Color Adjustments:** Intensify warmth for a fiery desert effect by significantly enhancing vibrance and saturation.
- **Individual Color Adjustments:** Amplify reds and yellows for a glowing rock formation. Boost the luminance and saturation of blues to create a vibrant, high-contrast sky against the landscape.

## Final Retouching Plan

### Step 1:  Light Adjustments

- Significantly increase **exposure** (+1.5 or more) to illuminate the overall image, countering the dark surroundings.
- Drastically reduce **highlights** and **whites** to recover sky details and prevent overexposure.
- Enhance **shadows** and **blacks** to bring out rich details in the foreground and rock formations.
- Increase **contrast** moderately to highlight the textures in the rock and terrain.

**Visual Effect:** The sky becomes more defined with visible cloud textures, and the landscape appears brighter and more detailed without losing depth.

### Step 2:  Color Adjustments

- Slightly warm the temperature to enhance earthy tones while ensuring the sky remains natural and inviting.
- Increase vibrance to bring out the richness of the desert colors, while keeping saturation subtle for authenticity.

**Visual Effect:** The reddish hues of the rock formations are more vibrant, complementing the blue tones in the sky for a balanced result.

### Step 3:  Individual HSL Adjustments

- **Reds and Oranges:** Enhance saturation and luminance slightly to make the rock formations stand out with more depth and texture.
- **Yellows:** Boost luminance to highlight the sunlit portions of the landscape.
- **Blues:** Brighten and slightly boost saturation to bring out the sky's clarity and create a subtle contrast with the rocks.
- **Greens:** Slightly increase luminance to emphasize vegetation details without overpowering the composition.

**Visual Effect:** The rocks take on a lively, warm tone while the sky and scattered vegetation add harmony without overpowering the natural feel.

### Summary

This approach balances the exposure changes, recovers sky details, and subtly enhances colors to achieve a natural, polished appearance. Each adjustment enhances the scene's realism while maintaining its authenticity.

## Histogram Analysis (iteration 1)

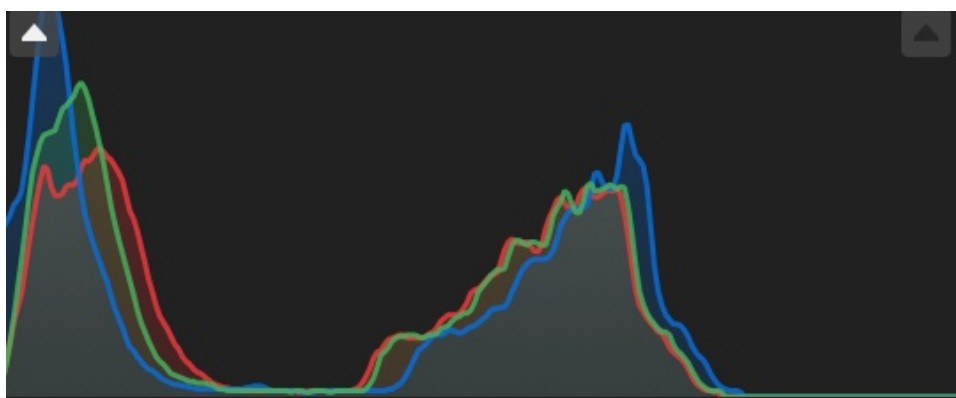

Histogram of the input image

The histogram shows a well-distributed tonal range but with potential areas for improvement. There is a moderate clustering on the far-left indicating adequately defined blacks, although a slight amount of clipping in the shadows is visible, which might cause some loss of detail in the darkest areas. The highlights and whites, represented by the far-right side, appear to be moderately pronounced but not excessively clipped, suggesting that the image retains most highlight details with slight room for brightening in the whites.

The midtones occupy the central portion of the histogram and represent a significant portion of the tonal range, showing a balanced exposure overall. The image likely has a mix of bright and dark elements with a mild leaning towards a low-key aesthetic, which may enhance mood but could also benefit from a slight lightening overall.

olor channels (red, green, and blue) exhibit alignment in mid and highlight regions, indicating good overall white balance and no significant color cast. However, the peaks in the red channel, particularly in shadow areas, may indicate a slightly warm tone in darker regions. The blue channel has a more distinct peak in highlights, suggesting cooler tones dominate in brighter areas of the image. Balancing these variations could bring a more harmonious and natural feel.

Overall, the histogram suggests a balanced image; however, minor adjustments such as reducing shadow clipping, enhancing midtone brightness, and ensuring harmony across color channels would elevate its visual appeal while retaining detail.

## Image Adjustment Parameters (Iteration 1)

### Basic Adjustments

| Parameter | Value |
|---|---|
| Exposure | +1.5 |
| Contrast | +20 |
| Highlights | −60 |
| Shadows | +50 |
| Whites | −20 |
| Blacks | +30 |
| Temp (Temperature) | 7000 |
| Tint | +10 |
| Vibrance | +30 |
| Saturation | +10 |

**Color Adjustments (HSL)**

| Color | Hue | Saturation | Luminance |
|---|---|---|---|
| Red | 0 | +15 | +10 |
| Orange | 0 | +20 | +10 |
| Yellow | 0 | +10 | +10 |
| Green | 0 | 0 | +5 |
| Blue | 0 | +10 | +15 |

The image needs significant brightening and saturation to bring out warm, earthy tones of the desert and details in the cloudy sky. Adjustments aim to balance light and color while ensuring the image maintains natural aesthetics.

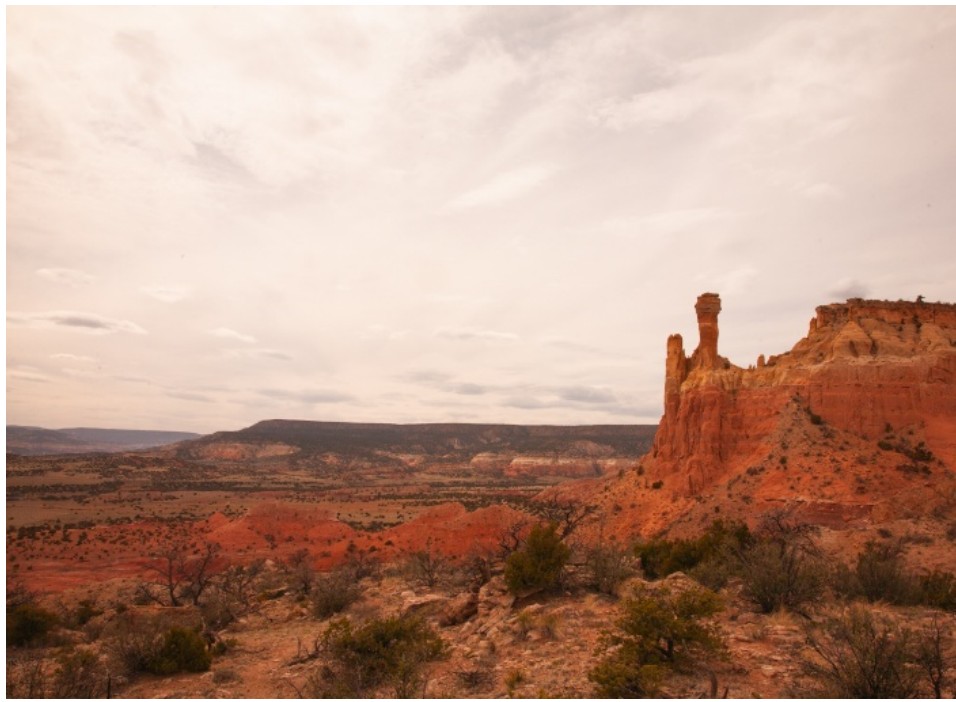

Retouched image (iteration 1)

**Reflection (iteration 1)**

**The adjustment is NOT satisfactory.** The current adjustment needs improvement. While the image appears brighter and highlights the landscape, shadow clipping occurs in darker areas, reducing detail as seen on the histogram's left side. Whites and the warm tones are overemphasized, leading to color imbalances and deviating from the natural aesthetic user preference. Suggest reducing reds and yellows, improving blue tones, softening blacks, and increasing midtone illumination.

## Histogram Analysis (iteration 2)

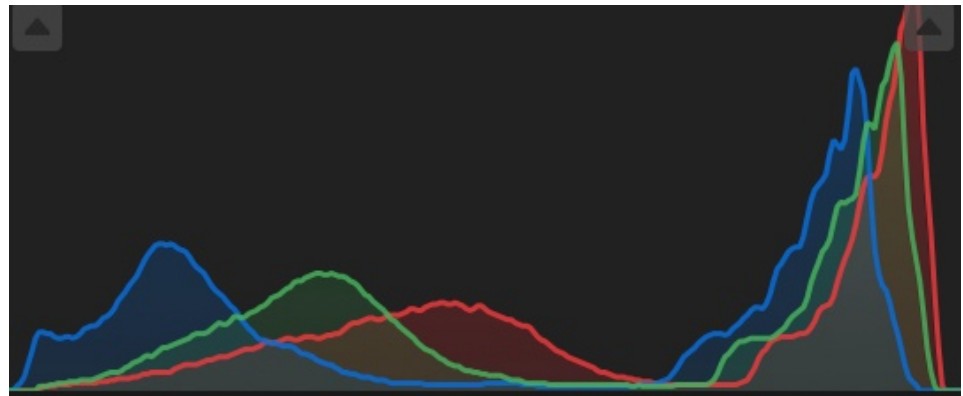

Histogram of the retouched image (iteration 1)

The histogram reveals a distinct tonal distribution, crucial for adjusting the image's light and color. Starting with **exposure and tonal range**, there is substantial data concentrated on the brighter end, particularly in the highlights and whites, as evident from the right-edge clustering. This prominence suggests an overall bright image with minimal underexposure issues. However, slight upward clipping in the bright areas points to a potential loss of detail in highlights, which, if unmanaged, could diminish subtle textures. At the opposite end, the shadows and blacks show a controlled presence, indicating no black clipping and a preserved shadow detail, underpinning the image's overall depth.

In terms of **white balance (temperature and tint)**, clear channel imbalances are visible: the red and green channels peak more substantially than the blue channel, particularly in the midtone and highlight ranges. This dominance implies a warmer tone within the image, likely skewing toward yellows or earthy reds. The blue channel, comparatively subdued, highlights a deficiency in cooler tones, which could disturb color harmony if not intended.

Other noteworthy features include the lack of sharp spikes or erratic gaps, pointing to a naturally graduated tonal distribution. Still, the histogram demonstrates inefficiencies in balancing cool and warm tones, which could be remedied while preserving overall depth and tonal structure. Overall, the image stands well in brightness and dynamic range but could benefit from rebalancing for color neutrality and slightly adjusting highlight preservation for improved clarity and texture retention.

## Image Adjustment Parameters (Iteration 2)

**Basic Adjustments**

| Parameter | Value |
|---|---|
| Exposure | +1.5 |
| Contrast | +20 → +25 |
| Highlights | −60 → −50 |
| Shadows | +50 → +40 |
| Whites | −20 |
| Blacks | +30 → +20 |
| Temp (Temperature) | 7000 → 5500 |
| Tint | +10 |
| Vibrance | +30 |
| Saturation | +10 |

**Color Adjustments (HSL)**

| Color | Hue | Saturation | Luminance |
|---|---|---|---|
| Red | 0 | +15 | +10 |
| Orange | 0 | +20 | +10 → +15 |
| Yellow | 0 | +10 | +10 → +20 |
| Green | 0 | 0 → +5 | +5 → +15 |
| Blue | 0 | +10 | +15 → +20 |

This approach aligns with the strategy to brighten the image, recover detail in the sky, and enrich the natural tones of the desert scene. Adjustments maintain the earthy warmth while enhancing vibrance and clarity, carefully balancing the dominant reds and yellows with cooler blues for cohesion.

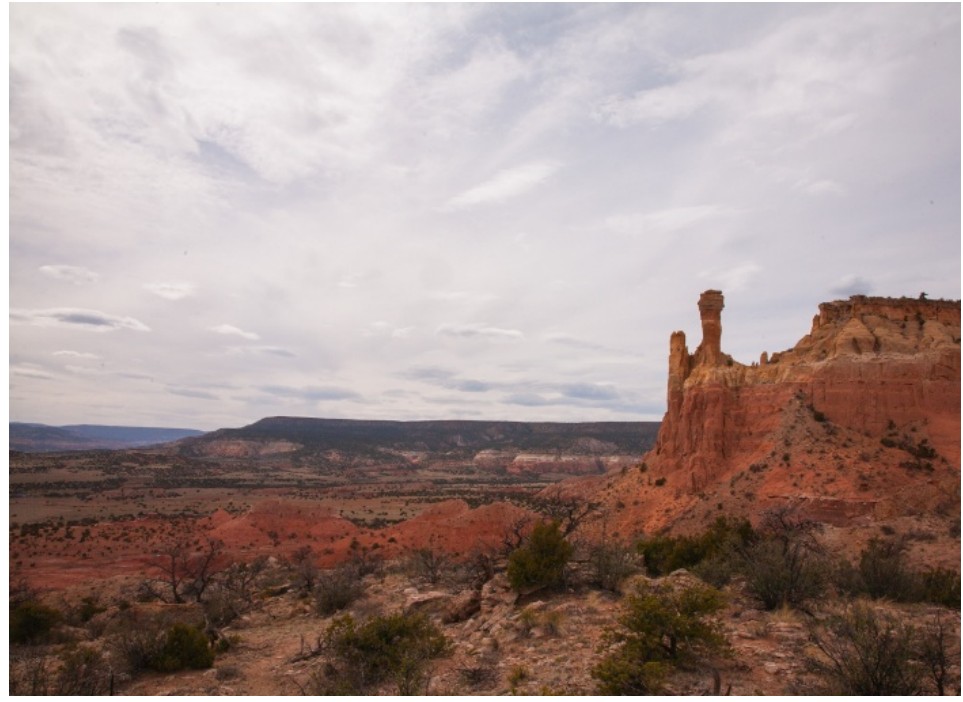

Retouched image (iteration 2)

## Reflection (iteration 2)

**The adjustment is NOT satisfactory.** The adjustments need improvement. The histogram reveals that highlight clipping persists, leading to loss of detail in the brightest areas, particularly in the sky. Additionally, the warmed tone dominates too strongly due to the subdued blue channel, disrupting color harmony. Recommend further reducing highlights, increasing black depth for contrast, and correcting the blue channel balance to better align with the intent.

## Histogram Analysis (iteration 3)

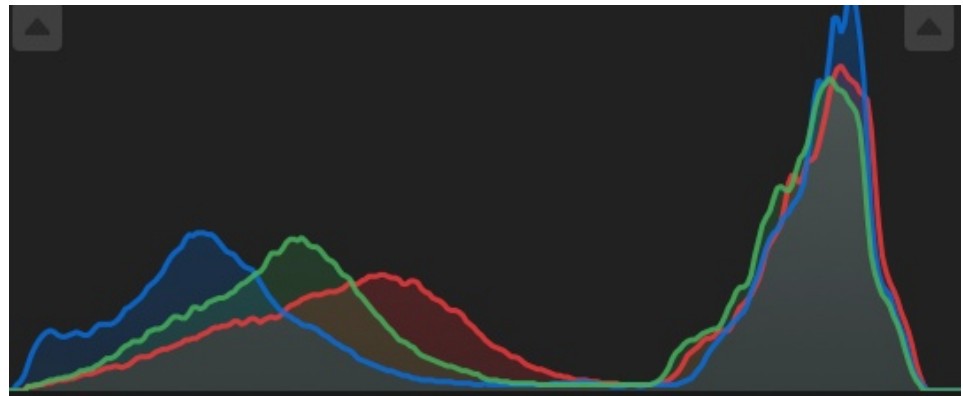

Histogram of the retouched image (iteration 2)

The histogram reveals a tonal distribution significantly skewed toward the highlights, indicating an image with a high-key leaning and prominent bright tones. There is an evident clustering toward the far-right edge in the histogram, particularly in the blue channel, suggesting possible highlight clipping, which risks losing detail in the bright areas. The left section shows no significant data clipping, meaning the shadows and true blacks are preserved, but there might be an opportunity to deepen blacks slightly for enhanced depth and contrast.

The RGB channels reveal imbalances, with the red channel peaking more prominently across the midtones and highlights, followed by green, while the blue channel is slightly subdued. This results in an overall warmer tonal bias, leaning toward yellows and diminishing the balance with cooler tones. The overlapping spikes in the highlights for all RGB channels indicate contributions to brightness and a clustered tonal range but may need further refinement to prevent oversaturation or excessive warmth.

Overall, the histogram suggests a bright and warm image, albeit with limited tonal depth and potential oversaturation in highlights. Enhancing shadow depth, balancing midtones, and adjusting the blue channel to cool down the warmth slightly could ensure a visually cohesive and balanced outcome.

## Image Adjustment Parameters (Iteration 3)

**Basic Adjustments**

| Parameter | Value |
|---|---|
| Exposure | +1.5 → +1.2 |
| Contrast | +25 → +20 |
| Highlights | −50 → −60 |
| Shadows | +40 |
| Whites | −20 → −30 |
| Blacks | +20 → −20 |
| Temp (Temperature) | 5500 → 5700 |
| Tint | +10 → 0 |
| Vibrance | +30 |
| Saturation | +10 → +15 |

**Color Adjustments (HSL)**

| Color | Hue | Saturation | Luminance |
|---|---|---|---|
| Red | 0 | +15 | +10 |
| Orange | 0 | +20 → +10 | +15 |
| Yellow | 0 | +10 | +20 |
| Green | 0 | +5 | +15 → +10 |
| Blue | 0 | +10 | +20 |

The adjustments aim to brighten the foreground, recover sky details, and slightly warm the image while retaining natural vibrancy. Individual hues and luminance are refined to emphasize the rocks, deepen sky tones, and subtly enhance vegetation to align with the original desert mood.

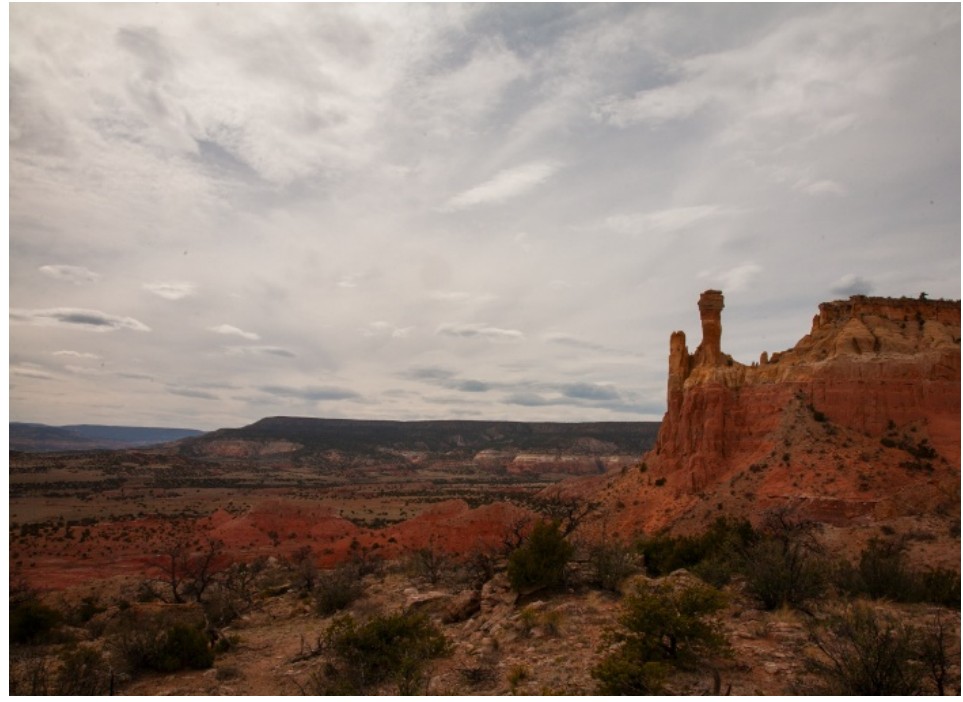

Retouched image (iteration 3)

## Reflection (iteration 3)

**The adjustment is satisfactory.** The adjustments have enhanced the image significantly by balancing the tonal range, recovering sky details, and brightening the foreground. Shadows and blacks have added depth without overpowering vibrancy, while the enhancements to the blue channel resolved the slight tonal imbalance, creating a harmonious and realistic desert aesthetic.

