# OpenReview forum: "PhotoArtAgent: Intelligent Photo Retouching with Language Model-based Artist Agent"
_ICLR.cc/2026/Conference — ICLR 2026 Conference Withdrawn Submission_

### Official Review · Reviewer_e4Tv · 2025-10-29

**Soundness:** 2
**Presentation:** 2
**Contribution:** 1
**Rating:** 2
**Confidence:** 4

**Summary:**

The paper proposes an VLM based system for photo editing. The editing mechanism relies on Adobe Lightroom to apply the edit. In particular, given an input photo, and a user request of a text edit, the VLM outputs the Lightroom editing parameters that can match the user's request. As an additional step, the authors allow re-processing the edited image with a VLM to evaluate the edit quality, and iteratively edit the photo until it reaches a satisfactory state. Evaluation is done through a user study, comparing the VLM based methods to humans and existing baselines.

**Strengths:**

- The method uses existing VLMs without any fine-tuning requirements, making it easy to implement
- Showcases that existing VLMs/LLMs can produce image editing parameters that are reasonable
- The method allows using a reference image as a guideline for the edit, which can be easier for the user to provide instead of providing a detailed description of the desired edit

**Weaknesses:**

- The logic of the paper is: There are two paradigms for editing. 1- direct editing with text-based models like Google Nano-Banana for example, or 2- editing in an interpretable way by producing editing parameters that can be used with off-the-shelf software like Lightroom. The argument of the authors is, (2) is preferred as it can be interpreted easily by the user. But the premise of the paper is editing for users with no editing background, and in that case, providing them with interpretable parameters is not very meaningful. So logically, it seems like choice 1 (direct editing) is more expressive and would be capable of producing better results.

- The fact that the VLM approach was preferred to users over human "experts" is questionable. Why did the humans perform not as well? were they too time restricted?

- The standard Lightroom auto is highly competitive with the proposed method in the user study, despite being significantly faster and cheaper. It is not clear if the cost and runtime of the method is justifiable.

- The user studies evaluate the quality of the edit, but there is no experiment evaluating whether the text edit provided by the user matches the produced output or not.

**Questions:**

- How are the human experts who were used as a baseline were selected? What credentials did they have? Also it seems that they are labeled as Expert A and Expert C, which makes me wonder was "Expert B" simply not included in the comparison?

- Is the contribution of the paper effectively a "prompt" for VLMs? It is unclear to me why a prompt is a sufficient contribution for a full research paper

- I can see the paper to be a plausible feature added in Lightroom or an editing software, but what makes it valuable to the research community? That remains unclear to me

- How is the communication between the VLM and Lightroom implemented? and why does it take at least a minute for the communication to occur? Does that mean that in cases where the model needs to re-iterate on the edit, the runtime can be as high as 14 minutes?

---

### Official Review · Reviewer_ThQY · 2025-10-30

**Soundness:** 3
**Presentation:** 4
**Contribution:** 2
**Rating:** 4
**Confidence:** 3

**Summary:**

This paper presents PhotoArtAgent, an intelligent photo retouching system that employs large Vision-Language Models to analyze images and emulate a human artist’s workflow. The agent explicitly reasons about artistic intent, proposes retouching strategies, generates concrete Lightroom parameters via an API, applies these parameters, and iteratively refines the results while providing reflective, text-based explanations. The workflow supports multimodal user input (text, reference images, retouching cases) and operates without model training. In both user studies and automatic evaluations, PhotoArtAgent matches or surpasses state-of-the-art automated tools and, in some cases, professional human artists.

**Strengths:**

1. This paper introduces an agentic, cognitive workflow for photo retouching based on VLM reasoning.
2. It also proposes a modular system supporting iterative reflection and multimodal user interaction.
3. This paper demonstrates parameter-level, interpretable editing via direct control of Lightroom.
4. Through quantitative and qualitative results, it shows that the proposed method achieve comparable results with human experts and deep-learning models.

**Weaknesses:**

1. From my perspective, this paper lacks sufficient novelty with a simple agentic framework for photo retouching.
2. The histogram based analysis may not be generalizable, also this paper heavily rely on Lightroom-style global adjustment, limiting its application.
3. Aesthetic evaluations based on VLM may not align with human perception.

**Questions:**

1. How to deal with the hallucination in VLM.
2. How to align VLM-based aesthetic evaluation with human value.
3. Can your agentic workflow generalized to other software and deep-learning based methods.

---

### Official Review · Reviewer_u4LX · 2025-10-31

**Soundness:** 2
**Presentation:** 3
**Contribution:** 1
**Rating:** 2
**Confidence:** 4

**Summary:**

PhotoArtAgent is an end-to-end image retouching system that uses top-performing LLMs to itertaively suggest image editing instructions, which are fed to image editing tools such as Lightroom. The entire pipeline of image analysis, initial proposal and iterative refinement mimicks human professionals. Using this system, the authors match human experts in an extensive user study.

**Strengths:**

1. Core technical details, such as detailed prompts, are provided in the paper, making it a good contribution for practitioners.

2. This work provides an end-to-end system, a good engineering effort.

3. There is a justification for iterative refinement, as 76.2% of the edited images required multiple iterations(fig 5b).

**Weaknesses:**

Major
1. Wrong venue for the contribution made: This work appears to be more of a systems demonstration than a meaningful submission to the ICLR main conference. There are no advances made or indicated in the paper, in image retouching, evaluation of image aesthetics, reasoning or general abilities of LMMs. L197("all these steps are performed by the same VLM with different prompts") effecitvety indicate prompt engineering, which cannot be a scientific contribution.

2. PhotoArtAgent is entirely training-free, effectively just an API call to top LLMs. The work can be condensed into:  LMMs with sufficient prompt engineering + Lightroom API can do photo retouching.

3. There are very limited technical contribution compared to MonetGPT[1], both of which perform the same task. While PhotoArtAgent proposes a system engineering + prompt design framework, MonetGPT has a novel training pipeline. This work depends on commerical APIs while MonetGPT is reproducible. This work incurs continuous costs through API calls while MonetGPT has a one-time training cost. This work relies on a single user study as evaluation whereas MonetGPT is tested on many benchmarks, with considerable ablations. While iterative refinement with reasoning is done by this work, it is already well studied in NLP.

4. There is no engagement with Image Quality Assesment works, which is an essential baseline to have. For example, testing LMMs on Q-Bench would indicate which one to finally choose.

Minor
1. The submission is formatted to ICLR 2025. While not grounds for desk rejection, the formatting should be fixed.

[1] Dutt et al. MonetGPT: Solving Puzzles Enhances MLLMs’ Image Retouching Skills, SIGGRAPH 2025

**Questions:**

1. Is 0 (noticeable flaws) to 10 (creative and satisfying) the best set of proxies to use? One checks technical quality and the other artisitic creativitiy.

2. This work uses GPT-4V to judge the aesthetic quality of suggestions made by another GPT4 model (GPT4-o). Is this reliable? GPT models will have inherent aesthetic preferences within themselves.

3. Is there an intuitive reason why a separate histogram benefits the final result, when this information is already inferrable by the VLM from the image itself?

---

### Official Review · Reviewer_9PLW · 2025-10-31

**Soundness:** 2
**Presentation:** 3
**Contribution:** 2
**Rating:** 2
**Confidence:** 4

**Summary:**

This paper proposes PhotoArtAgent, an agent that uses existing multimodal large language models such as GPT-4o to understand and edit photos through interaction with users. The agent adjusts photo parameters in Adobe Lightroom, including color, lighting, and tone, according to user requests. The authors evaluate the system through user studies and GPT-4V evaluation scores, with the main experiments conducted on the MIT-Adobe FiveK dataset.

**Strengths:**

1. The paper is clearly written and well explained. The figures effectively illustrate the workflow and are visually appealing, showing a good graphical style. Overall, it is easy to understand.

2. Photo editing itself is an interesting task, and using powerful large language models to control professional tools such as Lightroom for editing is an appealing and interesting application direction.

**Weaknesses:**

This topic itself is quite interesting, but the current submission has several issues:

1. The paper’s GPT-based agent and the designed workflow, in the context of today’s agent research, show limited insight therefore the contribution is relatively limited. The overall agent and workflow design are quite straightforward. The explanation and reflection mechanisms are common in current agent or workflow studies.

2. Using Lightroom (Lr) for editing with agents is an interesting task. However, if the highlights in agent or workflow design are relatively weak, I would expect some unique findings in this specific setting (Lr agent for editing) compared to other popular agent application scenarios. Yet, in Section 3, the analysis remains rather general. For example, the subsection on Effects of the Reflection Mechanism is a bit trivial, in most cases, we would see better results with multiple intuitive trials/reflections.

3. Experimental setup. I noticed that the parameters adjusted include light, tone (temperature, tint, etc.), and color. These are relatively simple and fundamental parameters, available in almost all mainstream editing apps, and default phone editing tools. On one hand, this shows that these parameters are fundamental; on the other hand, it also means the editing tasks are quite basic. Some examples in the paper (e.g., Fig. 7 and 8) and those in the MIT-Adobe FiveK dataset are actually simple, mostly involving global technical adjustments (such as exposure and color correction), without expressive or stylized editing, or local editing. This also explains why in Table 1, user study shows that, Lightroom Auto performs quite well. This largely reduces the contribution of the paper. Has the author attempted more localized adjustments, such as using masks, gradient filters? Or some more practical adjustments like curve adjustment? That would make the work more interesting and practical, given that authors choose the powerful Lightroom as the tool.

**Questions:**

1. Most adjustment in MIT-Adobe FiveK are quite simple, related to global technical adjustments (such as exposure and color correction). Lightroom Auto is good at such adjustments (Figure 7 shows comparisons. PhotoArtAgent seems to involve overexposure in certain areas ). Given that authors showed some stylized editing examples, would you consider doing some related evaluation?

2. If we just consider the basic adjustments related to technical correlation, like the main evaluation in the submission on MIT-Adove FiveK, I wonder if PhotoArtAgent can show obvious advantages over Lightroom Auto? The LLM based workflow can be very time-inefficient, considering CoT, reflection, etc. While Lightroom Auto is done in one second. In this case, I wonder how authors would explain the advantage of PhotoArtAgent? From user study and examples in Fig. 7, they are very close, and in fig. 7, Lightroom Auto may be even a little better in some cases.

3. Have authors considered more editing? Given the powerful features in Lightroom, and the image understanding capibility of GPT, performing simple global edits in light/tone/color can be trivial. Have you tried local edits or curve edits, which are also practical and frequently used in post-processing?

Overall I think the topic is interesting. But the contributions of the current version is limited. Details are available in Weaknesses.

---

### Note · Authors · 2025-11-13

I have read and agree with the venue's withdrawal policy on behalf of myself and my co-authors.